# REFERENCES IMPROVE LLM ALIGNMENT IN NON-VERIFIABLE DOMAINS

**Kejian Shi**[*,1], **Yixin Liu**[*,1], **Peifeng Wang**[2]
**Alexander R. Fabbri**[3], **Shafiq Joty**[4,5], **Arman Cohan**[1]
[1]Yale University [2]Meta [3]Scale AI [4]Salesforce Research [5]Nanyang Technological University
kejian.shi@yale.edu, yixin.liu@yale.edu, arman.cohan@yale.edu

## ABSTRACT

While Reinforcement Learning with Verifiable Rewards (RLVR) has shown strong effectiveness in reasoning tasks, it cannot be directly applied to non-verifiable domains lacking ground-truth verifiers, such as LLM alignment. In this work, we investigate whether high-quality reference outputs can be effectively leveraged to bridge this gap. First, we design evaluation protocols that enhance LLM-based evaluators for LLM alignment using reference outputs. Through comprehensive experiments, we show that a reference-guided approach substantially improves the accuracy of less capable LLM-judges using references from frontier models; stronger LLM-judges can also be enhanced by human-written references. We then demonstrate the utility of high-quality references in alignment tuning, where LLMs guided with references are used as judges to self-improve. We show that reference-guided self-improvement yields clear gains over both SFT distillation and reference-free baselines, achieving performance comparable to training with finetuned reward models. Specifically, our method achieves scores of 73.1% and 58.7% on AlpacaEval and Arena-Hard with Llama-3-8B-Instruct, and 70.0% and 74.1% with Qwen2.5-7B. These results highlight the potential of using reference-guided LLM-evaluators to enable effective post-training in non-verifiable domains.

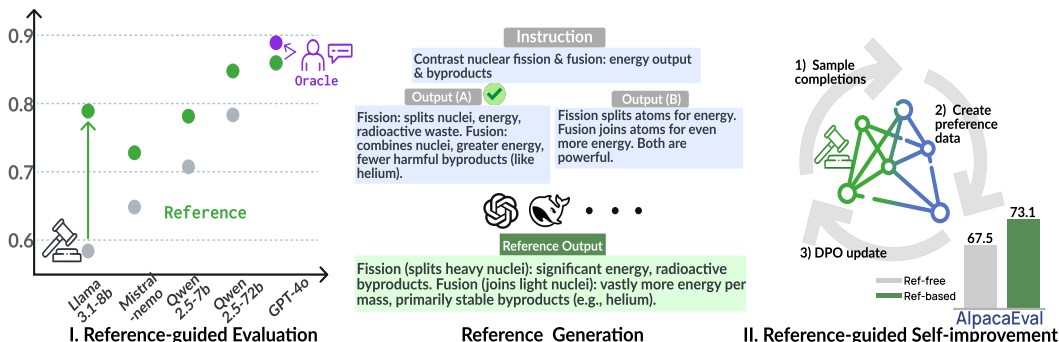

Figure 1: Overview of our study on reference-guided LLM-as-a-Judge for LLM alignment. Conceptual plots illustrating (I) the improvement in average accuracy from reference-guided evaluation (§3.3) and (II) the reference-guided self-improvement (§4).

# 1 INTRODUCTION

Recently, Reinforcement Learning (RL) from Verifiable Reward (RLVR) (Liu et al., 2024a; Lambert et al., 2025) has shown strong effectiveness in improving LLMs' reasoning capabilities. However, RLVR cannot be directly applied to non-verifiable domains, such as alignment tuning (Ouyang et al., 2022; Bai et al., 2022), because it is non-trivial to design verifiable/reliable rewards for these tasks. Consequently, RL from Human Feedback (RLHF) (Stiennon et al., 2020; Ouyang et al., 2022) or AI

---

[*]Equal Contribution.

Feedback(RLAIF) (Bai et al., 2022), remains the predominant paradigm for LLM post-training in these domains.

In this work, we propose to improve LLM alignment tuning by developing automatic evaluators that can effectively leverage high-quality reference outputs, which reduces the gap between RLVR and RLHF/RLAIF. Specifically, in RLHF/RLAIF, reward models or LLMs-as-Judges (Zheng et al., 2024; Li et al., 2023; 2024) are used as the automatic evaluators to provide the supervision/reward signals. These evaluators usually assess the model outputs in a reference-free manner. In contrast, various RLVR methods use "verifiers" to evaluate outputs against gold-standard solutions for tasks such as math reasoning (Liu et al., 2024a). This difference motivates us to explore whether high-quality references can serve as soft verifiers. We study: **can high-quality references support effective LLM alignment tuning in a self-improvement setting, without external human or AI supervision?** We argue that this question is important for two reasons. First, it mirrors the advantages of RLVR—learning from reference-based supervision—while extending them to non-verifiable domains where rule-based verifiers are infeasible. Second, in practice, reference outputs may be available even when human feedback is costly or unavailable, making it critical to enable model improvement directly from references.

To address this question, we develop LLM-judges[1] that can effectively leverage reference outputs to provide supervision signals for preference optimization algorithms such as DPO (Rafailov et al., 2023; Tunstall et al., 2024). Critically, these reference-guided LLM-judges are used in a self-improvement manner, where an LLM serves as the judge to supervise its own training process (Yuan et al., 2024; Wu et al., 2024), so no external human or AI feedback is required.[2]

In §3, we first develop effective reference-guided LLM-judges for alignment evaluation. Several recent studies have explored guiding LLM-judges using references (Zeng et al., 2024; Lyu et al., 2024; Zhang et al., 2025; Krumdick et al., 2025). However, their evaluation settings are limited in the types of tasks considered and the number of LLMs used as judges, and a more systematic and comprehensive investigation is lacking (further discussed in §2). To this end, we first introduce targeted prompting protocols designed to leverage strong references for alignment evaluations. We then conduct comprehensive evaluations for the developed reference-guided evaluation method based on the prompting protocols. Specifically, our proposed method achieves a 6.8% absolute improvement over the reference-free baseline evaluated across 11 LLM-judges using reference outputs generated by a stronger LLM, GPT-4o (Hurst et al., 2024) (§3.4). Furthermore, this improvement is generalizable when different frontier LLMs are used to provide references (§3.5), and frontier LLMs like GPT-4o can also be enhanced as judges when provided with high-quality human references (§3.6).

Having developed the LLM-judges that can effectively leverage references, we apply them in a self-improvement setting for alignment tuning (§4). Specifically, we use the instructions in the widely used UltraFeedback dataset (Cui et al., 2023) to fine-tune Llama-3-8B-Instruct (Meta AI, 2024) and Qwen2.5-7B (Yang et al., 2024), with high-quality references generated by DeepSeek-V3 (Liu et al., 2024a). We conduct a reference-focused training process involving two stages – (1) first performing distillation, i.e., supervised fine-tuning (SFT) on the reference outputs, (2) then applying the LLMs to be improved as reference-guided LLM-judges in further preference optimization using DPO.

The experimental results highlight the clear benefits of high-quality references: (1) At the first stage, the SFT distillation on reference outputs outperforms preference optimization (DPO) based on a finetuned reward model; (2) At the second stage, reference-guided self-improvement using DPO further improves upon the SFT baseline, and shows greater gains compared to reference-free self-improvement. Furthermore, DPO with a reference-guided self-LLM-judge achieves performance comparable to DPO using a finetuned reward model of the same parameter size, without requiring the additional human or AI feedback typically needed to train such reward models. On AlpacaEval (Li et al., 2023) and Arena-Hard (Li et al., 2024), our reference-guided self-improved models show superior performance, achieving scores of 73.1 and 58.7 with Llama-3, and 70.0 and 74.1 with Qwen2.5, respectively.

Our contributions are primarily twofold:

---

[1]We use "LLM-judge" to refer to an evaluation method that uses an LLM as the judge for brevity.

[2]Recent work proposes rubric-based rewards for RL in non-verifiable domains (Gunjal et al., 2025; Huang et al., 2025) to address RLVR's limitations. We note these methods complement ours and can be used jointly.

(1) We propose effective methods for enhancing LLM-judges with reference outputs in alignment evaluation, and demonstrate through comprehensive experiments that high-quality references can substantially improve LLM-judges' accuracy in alignment evaluation.

(2) We show that reference-guided LLM-judges are effective in semi-self-improvement settings for LLM alignment, providing empirical evidence that high-quality reference outputs can be leveraged for effective model training in non-verifiable domains with reference-guided automatic evaluators.[3]

## 2 RELATED WORK

**LLM-as-a-Judge.** Using powerful LLMs as automated evaluators (LLM-as-a-Judge) is a growing practice for scalable evaluation, especially in instruction-following tasks (Zheng et al., 2024; Li et al., 2023; Dubois et al., 2024). Benchmarks like MT-Bench and Arena-Hard (Zheng et al., 2024; Li et al., 2024) utilize strong LLMs (e.g., GPT-4) as judges, and this paradigm also supports training data annotation for preference optimization algorithms like DPO (Yuan et al., 2024; Rafailov et al., 2023). However, LLM judges are known to exhibit limitations such as positional and verbosity biases (Zheng et al., 2024; Zhu et al., 2023; Ye et al., 2024). Mitigation efforts include Chain-of-Thought (CoT) prompting (Wei et al., 2022), answer swapping (Shi et al., 2024), and developing more robust evaluation protocols (Zeng et al., 2024; Liu et al., 2024c). Our work builds on this by investigating reference-guided prompting to enhance LLM judge accuracy and robustness. Trivedi et al. (2024) explores improving judges via internal self-rationalization (Chain-of-Thought). Our work is orthogonal, focusing on *external grounding* via references to address knowledge gaps rather than reasoning gaps.

**The Role of References in LLM Evaluation.** Traditional NLG evaluation often relies on reference outputs (e.g., BLEU (Papineni et al., 2002), ROUGE (Lin, 2004)), but their role in LLM-as-a-Judge for alignment evaluation, where single ground-truth references are often insufficient, has been less explored. Recent work has begun revisiting references: LLMBar (Zeng et al., 2024) used prompts that guide LLM-judges to generate reference outputs before evaluation; HREF (Lyu et al., 2024) incorporated human-written responses and reported improved performance over reference-free methods. However, these studies are limited in both the number of LLMs evaluated and dataset scale. While Krumdick et al. (2025) focused on reference use for question answering, **RevisEval** (Zhang et al., 2025) proposed generating response-adapted references to improve evaluation accuracy. Unlike RevisEval, which focuses on dynamic reference modification for static evaluation, our work extends the setting to model training and demonstrates the benefits of reference-guided supervision in self-improvement scenarios. Furthermore, our work provides a more systematic and large-scale investigation into reference-guided LLM-judges, covering 5 datasets and 13 LLMs.

**Self-Improving LMs and Generative RMs.** LLM-judges have also been used in model training, particularly in self-improvement settings where an LLM supervises its own training (Yuan et al., 2024; Wu et al., 2024; Yasunaga et al., 2024). A related line of work explores Generative Reward Models (Zhang et al., 2024; Mahan et al., 2024), where LLMs serve as reward models in preference optimization. Recent studies show that general-purpose frontier LLMs can perform competitively with finetuned discriminative reward models in this setting (Zhou et al., 2025; Frick et al., 2025). Our work builds on this direction by studying reference-guided LLM-judges for self-improvement, making the setting more feasible by providing additional grounding through high-quality references.

## 3 DEVELOPING REFERENCE-GUIDED LLM-JUDGES

To enable effective use of references in improving LLM alignment evaluation, we first develop robust reference-guided evaluation methods for LLMs-as-Judges, and conduct comprehensive evaluations of them against strong baselines.

### 3.1 PRELIMINARY

LLM-judges for alignment evaluation typically perform pointwise scoring of a single output given an instruction (Zheng et al., 2024), or pairwise comparison of two outputs (Li et al., 2024). In this

---

[3]We release the codebase at https://github.com/yale-nlp/RLRR (**RL** from **R**eference-Guided **R**ewards).

study, we focus on the pairwise comparison setting, as it matches the annotation format of various high-quality human-labeled alignment datasets such as LLMBar (Zeng et al., 2024), and is directly applicable to preference optimization algorithms like DPO. While our primary analysis focuses on this pairwise setting, we also conducted experiments on pointwise scoring to ensure the robustness of our findings. We present these results in Appendix B, which confirms that reference-guided evaluation also improves performance in a pointwise scoring setting. To evaluate an LLM-judge, human annotations are typically used as ground truth. Specifically, in the pairwise comparison task, the LLM-judge's evaluation accuracy is measured by the proportion of instances where the LLM-judge selects the same preferred output as the human annotators.

## 3.2 REFERENCE-GUIDED PROMPTING FOR LLM-JUDGES

We introduce targeted prompting strategies designed to effectively leverage reference answers in the LLM-as-a-Judge paradigm. As a baseline, we first introduce a strong reference-free prompting method, which we refer to as **Ref-Free (Ours)**. This prompt is designed for direct pairwise comparison without relying on any external reference answer (its template is in Figure 9). Its general structure follows the base prompt proposed in Zeng et al. (2024). However, we design the prompt to specifically instruct the model to assess instruction-following quality along with other critical aspects such as factuality and verbosity. As results show, this method outperforms many existing baselines.

Building on the reference-based approach, we extend it to a reference-guided setting by instructing the LLM to assess which candidate output more closely aligns with the quality and content exemplified by the reference, while still addressing the original instruction. We refer to this method as **RefEval**. While prior work has proposed similar reference-guided prompting methods (Zeng et al., 2024; Lyu et al., 2024), our approach offers more explicit guidance on how the reference output should be used (See prompt design in Appendix A.2).

---

**RefEval**

**User Message:**
Decide which output is better at following the instruction.
An effective and factually correct Reference Output is provided to aid your evaluation. This Reference Output demonstrates successful instruction-following. Here are some aspects to consider:

1. Outputs should precisely follow the instruction. If an output contains unrelated information or does not complete each and all requirements in the instruction, it means that output does not precisely follow the instruction.
2. You should check for factual correctness and accuracy of outputs. If an output contains factual errors (especially with numbers), it should be considered lower quality. Compare the output against the Reference Output to verify if that output is factually correct.
3. Outputs should contain only a brief effective response without any verbose explanation, unless the instruction explicitly asks for an explanation.
4. Understand how the Reference Output properly delivers a helpful, accurate, and natural response, and then compare how closely an output matches this successful Reference Output.
5. Extraneous content in an output that goes beyond what is present in the Reference Output should be discouraged.
6. The order in which the outputs are presented to you should NOT affect your judgment.

Select which output, "Output (a)" or "Output (b)", is better at following the instruction. Your answer should ONLY contain: "Output (a)" or "Output (b)":

---

Figure 2: A snapshot of **RefEval** method.

We show a snapshot of our core prompt in Figure 2, while the full prompt template is provided in Figure 10 in Appendix H. As shown in §3, this emphasis on reference utilization leads to clear improvements over previous methods.

To further emphasize the role of references, we design an additional prompting method, **RefMatch** (Figure 11), which instructs the LLM-judge to act primarily as a semantic and stylistic matcher, determining which candidate output more closely resembles the reference. Specifically, the LLM is

Table 1: Average evaluation accuracy (%) across five datasets using 11 open-source models as judges. We report the 95% bootstrap confidence interval for the average performance. **\*** indicates the method is significantly worse than RefEval ($p < 0.05$). Dataset acronyms are: Natural (Nat), Adversarial (Adv), MTBench (MT), and InstruSum (Ins).

| Method | Nat | Adv | MT | Ins | HREF | Avg |
|---|---|---|---|---|---|---|
| LLMBar-Base | 83.1 | 61.7 | 74.6 | 70.2 | 72.0 | 72.3 (-1.4, +1.5) * |
| HREF-Base | 84.1 | 54.0 | 76.5 | 70.8 | 77.3 | 72.5 (-1.5, +1.6) * |
| CoT | 82.0 | 60.1 | 75.4 | 69.1 | 69.6 | 71.2 (-1.5, +1.6) * |
| Prepair | 81.7 | 71.5 | 72.6 | 68.8 | 75.2 | 74.0 (-1.3, +1.4) * |
| Self-Ref | 84.6 | 66.7 | 73.5 | 69.5 | 72.4 | 73.3 (-1.3, +1.3) * |
| Self-Metric-Ref | 85.5 | 67.4 | 75.0 | 70.5 | 74.4 | 74.6 (-1.3, +1.4) * |
| Ref-Free (Ours) | 83.4 | 67.8 | 70.9 | 71.0 | 75.4 | 73.7 (-1.3, +1.4) * |
| LLMBar-Ref | 85.5 | 66.3 | 74.5 | 70.7 | 72.8 | 74.0 (-1.3, +1.4) * |
| HREF-Ref | 85.3 | 62.3 | 76.5 | 70.8 | 79.2 | 74.8 (-1.3, +1.4) * |
| RefMatch | 84.6 | 74.1 | 76.3 | 72.9 | 80.4 | 77.7 (-1.6, +1.6) * |
| RefEval | **86.8** | **74.9** | **76.7** | **74.5** | **82.7** | **79.1 (-1.4, +1.5)** |

explicitly instructed with: "Your goal is to determine which output demonstrates closer similarity to the reference."

For comprehensiveness, we also explored several variants of these core methods. While these variants exhibited interesting characteristics in certain scenarios, our primary focus in the main paper will be on **RefEval** and **RefMatch** due to their consistent strong performance and clarity. Detailed descriptions and results for all explored variants are provided in Appendix A and Appendix H.

## 3.3 EVALUATION SETUP

**Evaluation Setting and Metric.** We use *evaluation accuracy* as the main metric, computed based on human annotations as ground truth in the pairwise comparison setting. To mitigate potential positional biases, where the LLM-judge might favor an output based on its presentation order (Park et al., 2024), all reported accuracies are averaged across **two evaluation passes with the order of candidate outputs swapped**. All LLM-judge evaluations use greedy decoding.

**Datasets.** We use five human-annotated datasets to ensure robust evaluation across varied instruction types and complexities in alignment evaluation. These datasets are: (1) **LLMBar-Natural** (Nat) and (2) **LLMBar-Adversarial** (Adv) (Zeng et al., 2024), which contain carefully curated instruction-following examples; (3) **MTBench** (MT) (Zheng et al., 2024), a benchmark for multi-turn conversational abilities; (4) **Instrusum** (Ins) (Liu et al., 2024b), focusing on instruction-controllable summarization; (5) **HREF** (Lyu et al., 2024), a benchmark with human-written reference responses for instruction following across diverse scenarios. Further details on dataset characteristics and processing are provided in Appendix D.

**Obtaining Reference Outputs** Most of the datasets we use do not provide human-written references. Therefore, to study reference-guided LLM-judges, we focus on a setting where a strong frontier LLM, GPT-4o, is used as an oracle to generate reference outputs, while less capable LLMs are used as judges, guided by the generated references.

**LLMs Evaluated as Judges** As GPT-4o is used as the oracle to provide references, we primarily evaluate 11 LLMs that are less capable than GPT-4o but represent a diverse range of model families and sizes: qwen-2.5-72b, Llama-3.1-70b, gemma-2-27b, qwen-2.5-14b, mistral-nemo, gemma-2-9b, Llama-3.1-8b, qwen-2.5-7b, Llama-3-8b, glm-4-9b, and mistral-7b-v0.3. We provide additional details in Appendix F.

**Baseline Evaluation Methods.** We compare our proposed reference-guided evaluation methods against the following established LLM-as-a-Judge prompting strategies. **LLMBar-Base**: The vanilla pairwise comparison approach from Zeng et al. (2024), where the LLM directly predicts the preferred

Table 2: Comparison of average evaluation accuracy across 11 LLM judges (averaged over five datasets). References for **RefEval** were generated by **GPT-4o**. Best performance for each model is bolded. `Base` denotes `LLMBar-Base`. Full statistical significance analysis is provided in Appendix A.1.

| Method | qwen-2.5 -72b | llama-3.1 -70b | gemma-2 -27b | qwen-2.5 -14b | mistral -nemo | gemma-2 -9b | glm-4 -9b | llama-3.1 -8b | qwen-2.5 -7b | llama-3 -8b | mistral-7b -v0.3 |
|---|---|---|---|---|---|---|---|---|---|---|---|
| Base | 79.4 | 85.2 | 82.3 | 81.5 | 65.6 | 80.8 | 71.8 | 65.0 | 73.5 | 60.1 | 47.0 |
| RefFree | 83.4 | 85.6 | 80.1 | **83.3** | 63.7 | 80.8 | 77.3 | 71.8 | 74.5 | 72.3 | 61.2 |
| RefEval | **84.6** | **85.9** | **84.9** | 82.4 | **73.2** | **85.7** | **79.5** | **79.4** | **77.4** | **77.5** | **69.6** |

output. **CoT**: Also from Zeng et al. (2024), this method prompts the LLM to provide a Chain-of-Thought explanation before making its final judgment. **Self-Ref**: Adapted from Zeng et al. (2024), the LLM first generates its own reference answer to the instruction, which is then used as context during the pairwise evaluation of candidate outputs. **Self-Metric-Ref**: Combines **Self-Ref** with an initial step where the LLM generates key evaluation metrics (aspects to consider) for the given instruction, as proposed by Zeng et al. (2024). **LLMBar-Ref**: it uses the same prompting method as Self-Ref, but uses the references generated by the oracle, GPT-4o, instead of self-generated references. **PrePAIR**: Proposed by Jeong et al. (2024), this protocol first elicits pointwise analysis for each candidate, identifying drawbacks, and then performs a pairwise comparison informed by these analyses. **HREF-Base**: The reference-free prompting method proposed by Lyu et al. (2024). **HREF-Ref**: The reference-based prompting method from Lyu et al. (2024), which incorporated a reference output into the prompt. The prompt templates of these methods are in Appendix H.

## 3.4 Overall Result Analysis

We first examine the average performance across our suite of 11 open-source LLM judges.

As shown in Table 1, **RefEval** achieves the highest average evaluation accuracy (79.1%). This significantly outperforms reference-free baselines such as **LLMBar-Base** (72.3%) and **CoT** (71.2%), as well as other reference-based methods, **HREF-Ref** (74.8%) and **LLMBar-Ref** (74.0%). Our other core reference-based method, **RefMatch**, also demonstrates strong performance (77.7%), ranking second overall. These results show the effectiveness of directly grounding LLM judgments with a strong reference via our proposed methods.

**Inter-Judge Agreement.** Beyond accuracy, we analyze whether references help different LLM-judges align with each other. As shown in Appendix C.2, **RefEval** substantially improves the average pairwise agreement between judges compared to the reference-free baseline (81.4% vs 76.6%), indicating that references provide a shared grounding that reduces variance in decision-making.

Table 2 illustrates the individual performance for the 11 LLM-judges. It shows that RefEval overall achieves higher or comparable accuracy, and that smaller, less capable models benefit more from the reference-guided evaluations. For instance, with Llama-3-8b, RefEval achieves an absolute improvement of approximately 17.4% over **LLMBar-Base**. While stronger models like qwen-2.5-72b also benefit (84.6% for **RefEval** vs. 79.4% for **LLMBar-Base**), the relative gains are more pronounced for models that initially struggle with reference-free evaluation. Providing a strong reference through **RefEval** effectively enables small LLMs to achieve evaluation quality closer to that of much larger ones. In §3.6, we present a case study showing that stronger LLM-judges can also be further enhanced with human references. Appendix A.5 provides a more detailed analysis regarding the LLM-judge's performance grouped by their sizes.

## 3.5 References from Various Frontier LLMs

Our primary experiments in §3.4 established the efficacy of **RefEval** and **RefMatch** using a single strong reference from GPT-4o. Therefore, we explore the impact of varying the source of this single reference and investigate strategies for leveraging multiple references. Specifically, we maintain the 11 open-source LLMs as judges to be evaluated, but generate reference outputs using four additional frontier LLMs here: Claude-3.5-Sonnet (Anthropic, 2024), Claude-3.7-Sonnet (Anthropic, 2025), Gemini-2.0-Flash (Google Cloud, 2025), DeepSeek-V3 (Liu et al., 2024a).

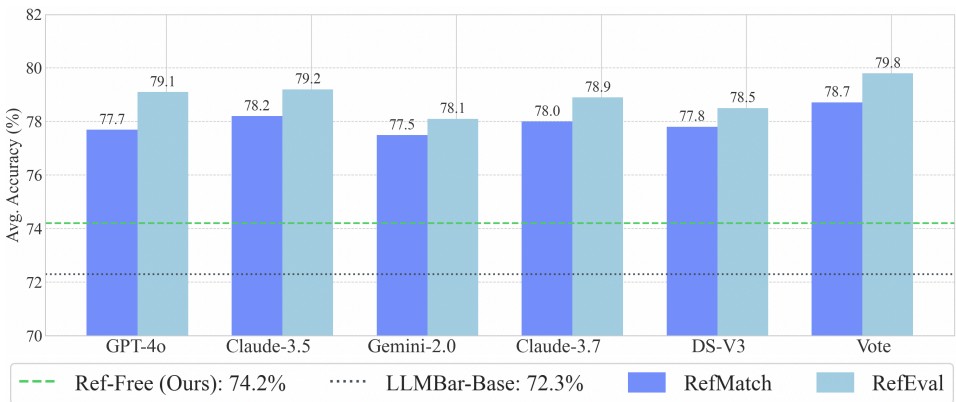

Figure 3: Evaluation accuracy of 11 open-source LLM-judges using **RefEval** and **RefMatch** with single references from various frontier models, and their voted versions. Horizontal dashed lines indicate reference-free baselines. Results are averaged over five datasets.

Figure 3 illustrates the performance of our **RefEval** and **RefMatch** methods when guided by a single reference from each of these frontier models. Both methods consistently outperform the reference-free baselines regardless of which frontier model generated the single reference, indicating their robustness.

Given the consistent benefit from different strong references, we then explored a **Multi-Reference Voting** strategy. Here, the LLM-judge performs independent pairwise evaluations for each candidate pair, each guided by a different reference. The final decision is made by a majority vote. As shown in Figure 3 (labeled *Vote*), this voting approach yields the highest average accuracies for both protocols.

**Robustness to Reference Quality.** While our primary experiments use frontier models as reference providers, we also investigated the impact of using references from less capable models. In an ablation study using **Llama-3.1-70B**

Table 3: Accuracy (%) on LLMBar-Adversarial comparing standard vs. human-edited "Oracle" references. Qwen is Qwen-2.5-72B and Llama is Llama-3.1-70B.

|  | GPT-4o | GPT-4.1 | Qwen | Llama |
|---|---|---|---|---|
| Ref-Free (Ours) | 85.4 | 85.3 | 71.0 | 80.3 |
| RefMatch | 84.2 | 86.1 | 81.0 | 79.9 |
| RefMatch-Oracle | 85.9 | 88.2 | **82.6** | 83.9 |
| RefEval | 86.8 | 86.7 | 79.9 | 82.8 |
| RefEval-Oracle | **88.4** | **88.6** | 81.8 | **84.6** |

as the judge, we replaced GPT-4o references with those generated by **Mistral-Nemo-12B** and **Tulu-2-7B**. As detailed in Appendix C.1, with these weaker references, RefEval achieved scores of 85.0% (Nemo) and 83.4% (Tulu), outperforming the reference-free baseline (80.3%).

## 3.6 HUMAN REFERENCES AS ORACLES

Our evaluation above uses GPT-4o to generate references to supervise less-capable LLM-judges. Therefore, to investigate the impact of high-quality references to stronger LLM-judges, we conduct a focused experiment on the **LLMBar-Adversarial** dataset, which is adversarially constructed and challenging to LLM-judges. To this end, we create "Oracle" references by having humans edit a subset of the machine-generated references (from GPT-4o) to ensure near-gold standard quality. Details of this editing process are in Appendix D.2. We evaluated four models, GPT-4o, GPT-4.1, Qwen-2.5-72B, and Llama-3.1-70B, using our RefEval and RefMatch protocols with both standard, sometimes flawed, machine references and these human-edited oracle references.

Table 3 shows that human-edited references enhance evaluation accuracy. For example, when GPT-4o serves as the judge, its **RefEval** accuracy increased from 86.8% with its self-generated reference to 88.4% with the Oracle GPT-4o reference. Similar improvements are observed across the models and protocols tested. These findings highlight that even highly capable LLM judges can benefit from

references of high, human-verified quality, particularly evident on adversarial or complex instructions where standard machine-generated references might contain flaws.

# 4 REFERENCE-GUIDED SELF-IMPROVEMENT

Having demonstrated the benefits of references in aiding LLM-judges' evaluations, we now explore their utility in model training. Specifically, we consider a self-improvement setting where an LLM supervises its own training using preference optimization algorithms (Yuan et al., 2024; Wu et al., 2024). Unlike prior work, however, the LLM is provided with high-quality reference outputs to guide its evaluations, making this setup more practical. This setting resembles classic NLG training, where reference outputs are available for supervised finetuning (SFT), but explicit preference annotations are absent. The reference-based (self-)LLM-judge offers a flexible alternative – extending the use of references beyond SFT and into preference optimization, without requiring preference-annotated data to train a separate reward model.

## 4.1 TRAINING PROCESS

**Training Stage 1: Distillation.** The first stage of the training process is *direct distillation*, where the base models are finetuned by SFT on the reference outputs. We found that this is superior to directly applying the preference optimization algorithms, which will be further discussed in §4.3.

**Training Stage 2: DPO.** At the second stage, the models are further finetuned using DPO (Rafailov et al., 2023):

$$\mathcal{L}_{\text{DPO}}(p_\theta; p_{\text{ref}}) = -\mathbb{E}_{(x,y_w,y_l)\sim D}\big[\log \sigma\big(\beta \log \frac{p_\theta(y_w|x)}{p_{\text{ref}}(y_w|x)} - \beta \log \frac{p_\theta(y_l|x)}{p_{\text{ref}}(y_l|x)}\big)\big], \quad (1)$$

where $x$ is an input in the training dataset $D$, $y_l$ and $y_w$ is a pair of outputs where $y_w$ is the better, and $\sigma(\cdot)$ is the sigmoid function. $p_\theta$ is the model under training, $p_{\text{ref}}$ is the reference model that is usually instantiated using the model checkpoint to be finetuned, and $\beta$ is a hyperparameter controlling the strength of the KL-divergence regularization from the reference model.

Here, the preference annotations are constructed "on-policy", where the output pair $(y_w, y_l)$ is sampled from the model to be finetuned, $p_{\text{ref}}$, at the beginning of the training, and annotated by an LLM-judge or a reward model. More specifically, we follow the setting of Meng et al. (2024) – sampling 5 candidate outputs for each instruction with a temperature of 0.8 and constructing the output pair by selecting the best and the worst candidates. When using LLM-judges that perform pairwise comparisons, all output pairs are compared to derive the average quality score of each candidate output. This process is adopted since previous studies (Dong et al., 2024; Meng et al., 2024) have found that such "on-policy" data generation methods lead to better performance compared to using static preference annotations, of which the output pairs are generated by different models.

## 4.2 EXPERIMENTAL SETTINGS

**Base Models.** We use two base models in our training experiments. The first model is **Meta-Llama-3-8B-Instruct**, which has already undergone post-training. To verify the effectiveness of reference-guided self-improvement on models that have not being substantially posttrained, the second model we use is **Qwen2.5-7B-SFT**, which we finetuned from the pretrained Qwen2.5-7B (Yang et al., 2024) on the Tulu3 SFT data mixture (Lambert et al., 2024a).[4] The SFT training setting follows the exact recipe used in Lambert et al. (2024a) (AppendixE).

**Evaluation Benchmarks.** We use two widely-used benchmarks for LLM alignment and instruction-following evaluation: AlpacaEval (Li et al., 2023) and Arena-Hard (Li et al., 2024). On both benchmarks, models' outputs are compared against GPT-4's outputs using a strong LLM as the judge. On AlpacaEval, we use the default LLM-judge, gpt-4-1106-preview. On Arena-Hard, we use gpt-4o-2024-08-06 as the judge, which is more cost-efficient and capable.

---

[4]https://huggingface.co/datasets/allenai/tulu-3-sft-mixture

Table 4: Performance comparison of training methods. Scores are for length-controlled AlpacaEval (AE) and Arena-Hard (AH).

| Method | Llama-3-8B-Instruct | | Qwen2.5-7B-SFT | |
| | AE | AH | AE | AH |
| --- | --- | --- | --- | --- |
| Base | 25.0 (-0.2, +0.3) | 27.1 (-1.8, 1.7) | 14.4 (-0.1, +0.2) | 23.4 (-2.1, 2.1) |
| ArmoRM-Base | 49.2 (-0.5, +0.5) | 40.4 (-2.4, 2.4) | 32.6 (-0.3, +0.3) | 58.6 (-2.3, 2.6) |
| V3-Distill | 53.9 (-0.5, +0.6) | 42.2 (-2.1, 2.3) | 48.8 (-0.5, +0.5) | 56.5 (-2.3, 2.1) |
| ROUGE | 56.4 (-0.6, +0.6) | 52.1 (-2.2, 2.5) | 50.9 (-0.5, +0.5) | 67.4 (-2.0, 2.5) |
| BERTScore | 58.8 (-0.6, +0.6) | 53.0 (-2.8, 1.9) | 55.3 (-0.5, +0.6) | 64.5 (-2.3, 2.6) |
| ArmoRM | **73.9** (-0.7, +0.7) | 58.6 (-0.7, +0.7) | 66.8 (-0.7, +0.7) | 72.2 (-2.2, 2.1) |
| RefFree | 67.5 (-0.7, +0.7) | 53.8 (-2.2, 2.6) | 65.1 (-0.6, +0.7) | 71.8 (-2.1, 2.1) |
| RefEval | 73.1 (-0.7, +0.7) | **58.7** (-2.8, 2.6) | **70.0** (-0.7, +0.7) | **74.1** (-2.4, 2.0) |

**Data Sources.** The instruction set used for preference optimization is from UltraFeedback (Cui et al., 2024), which consists of 60K instructions covering diverse scenarios. It has become a standard testbed for evaluating preference optimization algorithms (Tunstall et al., 2024; Meng et al., 2024). To obtain high-quality references, we use DeepSeek-V3 (Liu et al., 2024a) to generate outputs over the entire instruction set. DeepSeek-V3 is a frontier LLM that demonstrates strong capabilities across various domains, including competitive performance on AlpacaEval and Arena-Hard. Its strong performance and moderate API cost make it a suitable choice for reference generation.[5]

**Hyperparameters.** For the first training stage, SFT on the reference outputs, we use the same hyperparameter settings as those applied when finetuning Qwen2.5-7B on Tulu3 SFT as described above, following the recipe in Lambert et al. (2024a). For DPO training, we follow a similar setting as used in Meng et al. (2024). Specifically, the number of training epochs is 1, the batch size is 64, the maximum learning rate is 5e-7 with a cosine learning rate scheduler and 10% warmup steps. As for the hyperparameter $\beta$ in the DPO objective (Eq. 1), we perform a grid search within the range of $0.005 - 0.1$, and compare each algorithm using its best hyperparameter configuration, following evaluation settings in previous work (Tunstall et al., 2024; Meng et al., 2024).

### 4.3 RESULT ANALYSIS

Table 4 compares the performance of several models: (1) **Base** is the base model to be finetuned; (2) **ArmoRM-Base** applies DPO on the base model using the ArmoRM-Llama3-8B-v0.1 reward model (Wang et al., 2024), which achieves strong performance on RewardBench (Lambert et al., 2024b); and (3) **V3-Distill** is the SFT model distilled from DeepSeek v3 references. The following models are finetuned from **V3-Distill**: (4) **ROUGE**, which uses ROUGE scores as the reward[6]; (5) **BERTScore**, which uses the BERTScore metric (Zhang et al., 2020; Zhao et al., 2025); (6) **ArmoRM**, which uses the ArmoRM reward model; (7) **RefFree**, our reference-free self-improvement method (Figure 9); and (8) **RefEval**, our reference-guided self-improvement method (Figure 10), where the judges are the post-distilled Llama-3-8B-Instruct and Qwen2.5-7B-SFT.

Table 4 highlights the following findings:

(1) SFT training on high-quality reference outputs is more effective than performing preference optimization with a finetuned reward model. Specifically, V3-Distill, the distillation model, outperforms the model trained with ArmoRM. This underscores the benefits of strong reference outputs.

(2) LLMs can effectively self-improve, which is demonstrated by the substantial improvement of RefFree over V3-Distill.

(3) References help LLMs better self-improve, as RefEval, the model trained with the reference-guided self-judge, consistently outperforms RefFree. It also performs much more strongly than the traditional reference-based metrics, ROUGE and BERTScore, and achieves comparable or better

---

[5]The total cost of generating 60K reference outputs through DeepSeek's API is around 40 US dollars.

[6]Each output's quality score is the average of ROUGE-1 and ROUGE-2 scores against the reference outputs.

performance compared to the finetuned reward model ArmoRM. It shows that the LLM-judges' improvement from references observed in §3 can result in substantial improvement in training.

Table 5 provides a further comparison between the resulting model of our reference-based, self-improved training pipeline, and strong baselines. For Llama-3-8B-Instruct, we compare against SimPO (Meng et al., 2024), which is a competitive method that outperforms DPO. The compared checkpoint is trained from the same base model, on the same instruction set, following a similar training process, and uses ArmoRM as the reward model. For Qwen2.5-7B, we directly compare against its post-trained checkpoint, Qwen2.5-7B-Instruct (Yang et al., 2024). The results in Table 5 demonstrate a clear advantage of our trained models, validating the effectiveness of leveraging high-quality references.

Table 5: Performance of our self-improved models vs. strong baselines on AE and AH.

| | AlpacaEval | Arena-Hard |
|---|---|---|
| DeepSeek-V3 | 84.8 | 94.9 |
| SimPO-Llama3-8B-Inst | 51.6 | 36.2 |
| RefEval-Llama3-8B-Inst | 73.1 | 58.7 |
| Qwen2.5-7B-Inst | 29.9 | 58.0 |
| RefEval-Qwen2.5-7B | 70.0 | 74.1 |

**Impact of Reference Quality on Training.** To verify if the training benefits are exclusive to frontier references, we repeated the **RefEval** self-improvement pipeline using references generated by **GPT-4o-mini** (a weaker model than DeepSeek-V3). Detailed results are provided in Appendix C.1. We find that the resulting model still outperforms the reference-free self-improvement baseline using the same generator. This shows that while higher-quality references yield better results, the reference-guided supervision mechanism itself provides structural benefits regardless of the specific generator.

**Understanding the Benefits of Reference-Based Self-Improvement.** To better understand the distinction between reference-based and reference-free supervision, we use GPT-4o to categorize the instructions in AlpacaEval and Arena-Hard into four types: Coding&Math, Creative Tasks, Information Seeking, and Reasoning&Planning.

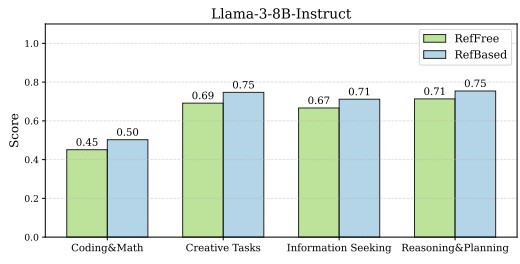

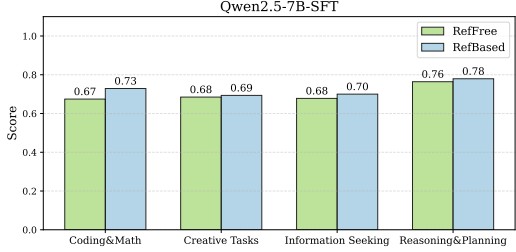

Figure 4: Comparison of reference-free and reference-guided self-improvement across task categories on AlpacaEval and Arena-Hard.

We then compare the performance of the RefFree and RefEval models across each category.[7] Figure 4 shows that for both Llama-3-8B-Instruct and Qwen2.5-7B-SFT, reference-based supervision yields a substantial improvement in the Coding&Math category. However, its benefit on Creative Tasks is less significant for Qwen2.5-7B-SFT, while remaining considerable for Llama-3-8B-Instruct. We posit that this is because leveraging references effectively in open-ended tasks is more challenging, and doing so requires more extensive post-training (as in Llama-3-8B-Instruct) rather than the standard SFT (as in Qwen2.5-7B-SFT).

## 5 CONCLUSION

In this study, we investigate whether high-quality reference outputs can enable effective LLM alignment tuning. Across five datasets, we show that high-quality references can consistently improve LLM-judge performance. Using the developed reference-guided LLM-judges in alignment tuning, we demonstrate that they can lead to effective semi-self-improvement by using high-quality references, even achieving performance comparable to that of trained reward models. Our findings highlight the potential of leveraging references to improve LLMs in non-verifiable domains, while reducing the methodology gap between RLHF/RLAIF and RLVR for LLM post-training.

---

[7]The prompt used for classification and the distribution of instruction types are in Appendix G.

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

## A  FULL EXPERIMENTAL RESULTS FOR EVALUATION PROTOCOLS

This appendix provides the complete evaluation accuracy results for all tested prompting protocols, including variants discussed in the main text as well as those explored during our broader investigation. All averages are computed over the five core datasets: LLMBar-Natural (Nat), LLMBar-Adversarial (Adv), MTBench (MT), Instrusum (Ins), and HREF. References for reference-based methods were generated by **GPT-4o** unless otherwise specified.

### A.1  FULL STATISTICAL ANALYSIS

Table 6 presents the detailed statistical analysis for individual LLM-judges, corresponding to the summarized results in Table 2 of the main text. We report the 95% bootstrap confidence intervals and paired significance tests. By transposing the view (Models as rows), we observe that **RefEval** provides a statistically significant improvement over the reference-free baselines for every single model tested.

Table 6: Detailed evaluation accuracy (%) by model with 95% Bootstrap Confidence Intervals. **\*** indicates the method is significantly worse than **RefEval** ($p < 0.05$). Comparison across 11 Open-Source Models. This table provides the full statistical context for Table 2.

| Judge Model | LLMBar-Base | Ref-Free (Ours) | RefEval |
|---|---|---|---|
| qwen-2.5-72b | 79.4 (-1.9, +1.8) * | 83.4 (-1.9, +1.8) * | **84.6** (-1.8, +1.8) |
| llama-3.1-70b | 85.2 (-1.8, +1.7) * | 85.6 (-1.7, +1.8) | **85.9** (-1.7, +1.8) |
| gemma-2-27b | 82.3 (-2.1, +2.0) * | 80.1 (-1.9, +2.0) * | **84.9** (-1.7, +1.8) |
| qwen-2.5-14b | 81.5 (-1.9, +1.9) * | **83.3** (-1.8, +1.9) * | 82.4 (-2.2, +2.1) |
| mistral-nemo | 65.6 (-2.2, +2.4) * | 63.7 (-2.2, +2.3) * | **73.2** (-2.1, +2.2) |
| gemma-2-9b | 80.8 (-1.8, +1.9) * | 80.8 (-2.2, +2.0) * | **85.7** (-2.0, +2.0) |
| glm-4-9b | 71.8 (-1.8, +2.1) * | 77.3 (-2.0, +2.0) * | **79.5** (-1.9, +1.9) |
| llama-3.1-8b | 65.0 (-2.4, +2.3) * | 71.8 (-2.1, +2.1) * | **79.4** (-2.2, +2.2) |
| qwen-2.5-7b | 73.5 (-1.9, +2.1) * | 74.5 (-2.0, +2.1) * | **77.4** (-2.0, +2.0) |
| llama-3-8b | 60.1 (-2.4, +2.5) * | 72.3 (-2.1, +2.1) * | **77.5** (-2.1, +2.3) |
| mistral-7b-v0.3 | 47.0 (-2.6, +2.6) * | 61.2 (-2.3, +2.4) * | **69.6** (-2.2, +2.4) |

## A.2 Prompt Protocol Design and Variants

We argue that many existing reference-guided approaches, such as **HREF-Ref** (Lyu et al., 2024) (Figure 13) or reference-augmented versions of LLMBar prompts (e.g., Figure 22) (Zeng et al., 2024), often treat the reference as supplementary information rather than a central anchor for judgment. These methods typically provide general evaluation criteria alongside the reference, without explicit instructions on *how* the reference should be actively used to ground the decision.

In contrast, our core protocols, **RefEval** and **RefMatch**, are designed to make the static reference a primary component of the evaluation process. **RefEval** explicitly instructs the LLM-judge to use the provided "Reference Output" as a benchmark for successful instruction-following, guiding it to compare candidates against this exemplar for factual correctness and overall quality (Figure 10).

**RefMatch** directs the judge to act as a semantic and stylistic matcher, determining which candidate shows closer similarity to the "ground-truth Reference Output" based on specific matching rules and an understanding of the reference's instruction-following pattern. Both protocols aim to provide more direct and robust guidance on leveraging the reference effectively (Figure 11).

For baseline comparisons, our **Ref-Free (Ours)** prompt (Figure 9) was developed as a strong reference-free method.

Table 7: Full average evaluation accuracy (%) across five datasets using **GPT-4o** as the judge, with references generated by **GPT-4o** itself.

| Method | Nat | Adv | MT | Instrusum | HREF | Avg |
|---|---|---|---|---|---|---|
| RefEval | 0.975 | 0.889 | 0.800 | 0.807 | 0.907 | 0.876 |
| RefEval-LLMBarRules | 0.970 | 0.865 | 0.820 | 0.803 | 0.899 | 0.871 |
| LLMBar-Base | 0.975 | 0.845 | 0.798 | 0.818 | 0.907 | 0.869 |
| Metric-Reference | 0.970 | 0.862 | 0.805 | 0.793 | 0.909 | 0.868 |
| Ref-Free (Ours) | 0.960 | 0.854 | 0.788 | 0.810 | 0.928 | 0.868 |
| Self-Reference | 0.960 | 0.853 | 0.805 | 0.793 | 0.906 | 0.863 |
| RefEval | 0.960 | 0.853 | 0.805 | 0.793 | 0.906 | 0.863 |
| Prepair | 0.965 | 0.865 | 0.788 | 0.799 | 0.876 | 0.859 |
| CoT | 0.975 | 0.832 | 0.800 | 0.788 | 0.854 | 0.850 |
| HREF-Ref | 0.945 | 0.801 | 0.812 | 0.794 | 0.888 | 0.848 |
| RefMatch | 0.925 | 0.842 | 0.760 | 0.813 | 0.888 | 0.846 |
| RefMatch-Rules-CoT | 0.960 | 0.851 | 0.800 | 0.755 | 0.835 | 0.840 |
| HREF-Base | 0.950 | 0.762 | 0.795 | 0.800 | 0.853 | 0.832 |

To further explore the design space, we also developed variants:

Table 8: Full average evaluation accuracy (%) across five datasets using 11 open-source models as judges, with references generated by **GPT-4o**.

| Method | Nat | Adv | MT | Instrusum | HREF | Avg |
|---|---|---|---|---|---|---|
| RefEval | 0.868 | 0.749 | 0.767 | 0.745 | 0.827 | **0.791** |
| RefMatch | 0.846 | 0.741 | 0.763 | 0.729 | 0.804 | 0.777 |
| HREF-Ref | 0.853 | 0.623 | 0.765 | 0.708 | 0.792 | 0.748 |
| RefMatch-Rules-CoT | 0.866 | 0.733 | 0.757 | 0.704 | 0.749 | 0.762 |
| RefEval-Rules | 0.862 | 0.672 | 0.760 | 0.718 | 0.749 | 0.752 |
| HREF-Base | 0.841 | 0.540 | 0.765 | 0.708 | 0.773 | 0.725 |
| Metric-Ref | 0.855 | 0.674 | 0.750 | 0.705 | 0.744 | 0.746 |
| Ref-Free (Ours) | 0.834 | 0.678 | 0.709 | 0.710 | 0.754 | 0.737 |
| Prepair | 0.817 | 0.715 | 0.726 | 0.688 | 0.752 | 0.740 |
| LLMBar-Ref | 0.855 | 0.663 | 0.745 | 0.707 | 0.728 | 0.740 |
| Self-Ref | 0.846 | 0.667 | 0.735 | 0.695 | 0.724 | 0.733 |
| LLMBar-Base | 0.831 | 0.617 | 0.746 | 0.702 | 0.720 | 0.723 |
| CoT | 0.820 | 0.601 | 0.754 | 0.691 | 0.696 | 0.712 |

- **RefEval-Rules** (Figure 17): Combines **RefEval** with a structured list of LLMBar-style evaluation rules, explicitly linking rule adherence to the reference.

- **RefMatch-Rules-CoT** (Figure 16): Augments **RefMatch** by requiring a Chain-of-Thought reasoning step before the final similarity judgment.

For multi-reference scenarios (Section 3.5), we explored:

- **Multi-Ref Avg** (Figure 14, §A.6): Asks the judge to consider overall similarity to a set of three references.

- **Multi-Ref MAX** (Figure 15, §A.6): Asks the judge to find the best match to any single reference within a set of three.

### A.3 PERFORMANCE WITH FRONTIER MODEL JUDGE

Table 7 presents the full evaluation accuracy across the five datasets for **GPT-4o** judge .

### A.4 OVERALL PERFORMANCE WITH OPEN-SOURCE LLM JUDGES

Table 8 provides the complete average performance of all prompting protocols when using the 11 open-source models as judges.

### A.5 PERFORMANCE BREAKDOWN BY OPEN-SOURCE MODEL SCALE

Tables 9 and 10 present the detailed performance breakdown for larger (¿9B parameters) and smaller (≤9B parameters) open-source model groups, respectively.

Figure 5 presents the aggregate performance of **LLMBar-Base**, **Ref-Free (Ours)**, and **RefEval** on each of the five datasets, segmented by model capability groups (Larger Models: ¿9B, including GPT-4o variants; Smaller Models: ≤9B).

For the larger model group (top panel), **RefEval** shows clear advantages on the challenging **LLMBar-Adversarial** and **HREF** datasets, while maintaining competitive performance elsewhere.

For the smaller model group, **RefEval** provides substantial gains, most notably on **LLMBar-Adversarial** (72.7% for **RefEval** vs. 56.8% for **LLMBar-Base**) and **HREF** (80.8% vs. 64.5%). This consistent improvement across diverse datasets reinforces the robustness and general applicability of the **RefEval** protocol, particularly its capacity to elevate the evaluation performance of resource-efficient smaller models.

Table 9: Full average evaluation accuracy (%) for larger open-source models (**Qwen-2.5-72B**, **Llama3.1-70B**) as judges, across five datasets. References by **GPT-4o**.

| Method | Nat | Adv | MT | Instrusum | HREF | **Avg** |
|---|---|---|---|---|---|---|
| RefEval | 0.915 | 0.813 | 0.795 | 0.797 | 0.878 | **0.840** |
| Metric-Ref | 0.935 | 0.790 | 0.825 | 0.773 | 0.857 | 0.836 |
| Self-Ref | 0.938 | 0.775 | 0.816 | 0.768 | 0.846 | 0.829 |
| RefMatch | 0.893 | 0.813 | 0.793 | 0.773 | 0.863 | 0.827 |
| Ref-Free (Ours) | 0.910 | 0.756 | 0.811 | 0.772 | 0.864 | 0.823 |
| LLMBar-Ref | 0.915 | 0.786 | 0.821 | 0.777 | 0.832 | 0.826 |
| HREF-Ref | 0.920 | 0.680 | 0.820 | 0.779 | 0.838 | 0.807 |
| LLMBar-Base | 0.905 | 0.735 | 0.824 | 0.768 | 0.835 | 0.813 |
| RefEval-Rules | 0.923 | 0.769 | 0.820 | 0.767 | 0.809 | 0.817 |
| Prepair | 0.933 | 0.827 | 0.785 | 0.769 | 0.807 | 0.824 |
| RefMatch-Rules-CoT | 0.942 | 0.812 | 0.819 | 0.748 | 0.785 | 0.821 |
| HREF-Base | 0.885 | 0.601 | 0.829 | 0.769 | 0.820 | 0.781 |
| CoT | 0.897 | 0.771 | 0.812 | 0.743 | 0.773 | 0.799 |

Table 10: Full average evaluation accuracy (%) for smaller open-source models (≤ 9B parameters) as judges, across five datasets. References by **GPT-4o**.

| Method | Nat | Adv | MT | Instrusum | HREF | **Avg** |
|---|---|---|---|---|---|---|
| RefEval | 0.845 | 0.727 | 0.759 | 0.715 | 0.808 | **0.771** |
| RefMatch | 0.817 | 0.701 | 0.751 | 0.701 | 0.759 | 0.746 |
| HREF-Ref | 0.828 | 0.580 | 0.741 | 0.684 | 0.759 | 0.718 |
| RefMatch-Rules-CoT | 0.837 | 0.696 | 0.735 | 0.684 | 0.722 | 0.735 |
| HREF-Base | 0.817 | 0.492 | 0.740 | 0.674 | 0.744 | 0.693 |
| Prepair | 0.784 | 0.678 | 0.698 | 0.646 | 0.725 | 0.706 |
| RefEval-Rules | 0.826 | 0.629 | 0.732 | 0.689 | 0.688 | 0.713 |
| Ref-Free (Ours) | 0.806 | 0.652 | 0.678 | 0.689 | 0.702 | 0.705 |
| Metric-Ref | 0.822 | 0.621 | 0.718 | 0.669 | 0.665 | 0.700 |
| LLMBar-Ref | 0.824 | 0.611 | 0.712 | 0.674 | 0.648 | 0.694 |
| CoT | 0.785 | 0.535 | 0.730 | 0.666 | 0.653 | 0.674 |
| Self-Ref | 0.805 | 0.621 | 0.703 | 0.656 | 0.641 | 0.685 |
| LLMBar-Base | 0.795 | 0.568 | 0.718 | 0.670 | 0.646 | 0.679 |

Table 11: Full average evaluation accuracy (%) with Multi-Reference Strategies using 11 open-source models as judges. "Vote" indicates Multi-Voting Aggregation. Methods without "Vote" (except Multi-Prompt) use a single GPT-4o reference unless they are inherently reference-free. Averages are over 4 datasets (Nat, Adv, MT, Ins).

| Method | Average Accuracy (%) on Datasets | | | | |
|---|---|---|---|---|---|
| | Nat. | Adv. | MT. | Ins. | **Avg.** |
| RefEval-Vote | 0.885 | 0.748 | 0.768 | 0.756 | **0.789** |
| RefEval (Single Ref) | 0.868 | 0.749 | 0.767 | 0.745 | 0.782 |
| RefMatch-Vote | 0.858 | 0.748 | 0.775 | 0.736 | 0.779 |
| RefMatch-Rules-CoT-Vote | 0.888 | 0.745 | 0.755 | 0.728 | 0.779 |
| RefMatch (Single Ref) | 0.846 | 0.741 | 0.763 | 0.729 | 0.770 |
| RefMatch-Rules-CoT (Single Ref) | 0.866 | 0.733 | 0.757 | 0.704 | 0.765 |
| Multi-Prompt-Avg | 0.857 | 0.674 | 0.781 | 0.736 | 0.762 |
| Multi-Prompt-Max | 0.863 | 0.681 | 0.748 | 0.702 | 0.748 |
| Metric-Ref (Single Ref) | 0.855 | 0.674 | 0.750 | 0.705 | 0.746 |
| HREF-Ref (Single Ref) | 0.853 | 0.623 | 0.765 | 0.708 | 0.737 |
| Prepair (OS Avg, Single Ref) | 0.817 | 0.715 | 0.726 | 0.688 | 0.737 |
| LLMBar-Base (OS Avg) | 0.831 | 0.617 | 0.746 | 0.702 | 0.724 |

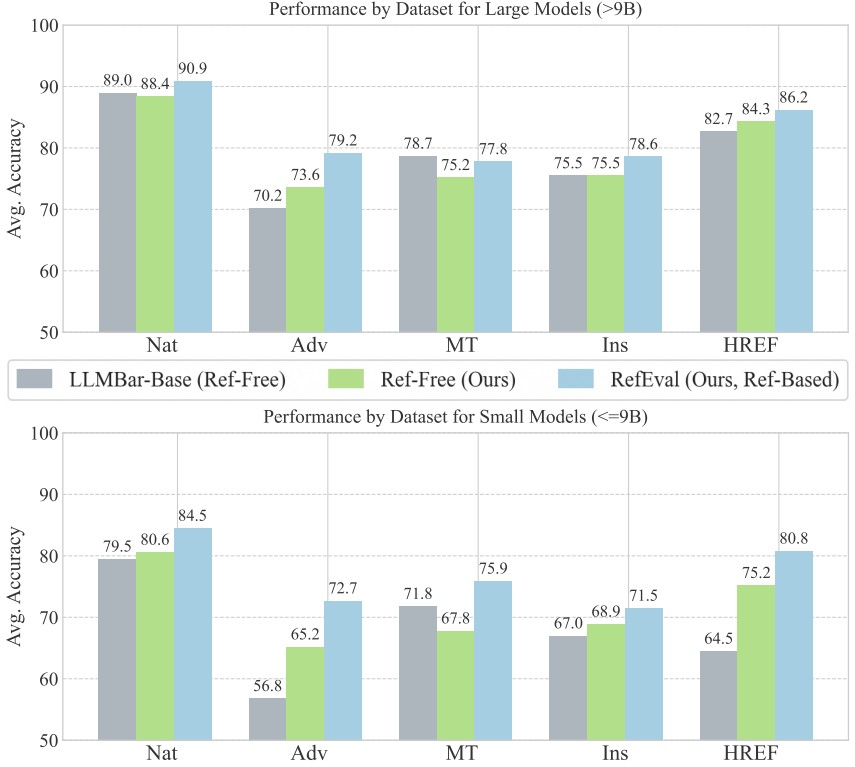

Figure 5: Aggregate performance by dataset for Larger Models (¿ 9B parameters, including GPT-4o variants; top panel) and Smaller Models (≤9B parameters; bottom panel). **RefEval** demonstrates consistent improvements across most datasets for both model groups.

## A.6 FULL RESULTS FOR MULTI-REFERENCE STRATEGIES

Table 11 presents the complete results for our exploration of multi-reference strategies, including Multi-Voting and Multi-Prompt variants, averaged across all 11 open-source LLM judges. The single-reference results for **RefEval** and **RefMatch** (using GPT-4o reference) are included for direct comparison.

## B ADDITIONAL RESULTS ON POINTWISE SCORING

Table 12: Evaluation accuracy (%) of pointwise scoring methods. Pointwise scores are used to infer pairwise preferences, which are then compared against human labels.

| Method | Nat | Adv | MT | Instrusum | HREF | Avg. |
|---|---|---|---|---|---|---|
| *Average of 11 Open-Source Models* | | | | | | |
| Base-point | 77.8 | 65.0 | 68.4 | 60.8 | 67.2 | 67.8 |
| RefEval-point | **82.7** | **71.7** | **73.1** | **67.4** | **73.1** | **73.6** |
| *Average of GPT-4o and GPT-4.1* | | | | | | |
| Base-point | **90.5** | 84.0 | 74.5 | 71.5 | 79.9 | 80.0 |
| RefEval-point | 89.5 | **87.1** | **75.2** | **72.6** | **83.9** | **81.7** |

We conducted additional experiments on pointwise scoring (§3.1). For this evaluation, we adapted our reference-based **RefEval** and the reference-free **LLMBar-Base** protocols to pointwise scoring formats, namely (**RefEval-point**) and (**Base-point**). In this setup, the LLM-judge is asked to rate a single model output on a Likert scale from 1 to 5 (see prompt at Figure 23 and Figure 24). To compute evaluation accuracy, the output with the higher score is designated as the winner. This

Table 13: Model registry and metadata described in §3.3. All models are post-trained.

| Name | Size | License | Description |
|---|---|---|---|
| gemma-2-9b | 9b | Gemma | Gemma is a family of open models from Google |
| gemma-2-27b | 27b | Gemma | (Gemma et al., 2024). These are instruct-tuned versions. |
| glm-4-9b | 9b | GLM-4 | GLM-4-9B is an open-source version of the latest generation of pre-trained models launched by Zhipu AI (Du et al., 2022). |
| llama-3-8b | 8b | llama 3 Community | llama 3 are the latest open models from Meta AI (Meta AI, 2024), pretrained on 15T tokens. |
| llama-3.1-8b | 8b | llama 3.1 Community | llama 3.1 collection offers a series of multilingual mod- |
| llama-3.1-70b | 70b | llama 3.1 Community | els that outperform many open and closed chat models on industry benchmarks (Llama3, 2024). |
| qwen-2.5-7b | 7b | Qianwen | Qwen is a family of models built by Alibaba Cloud (Bai |
| qwen-2.5-14b | 14b | Qianwen | et al., 2023). Qwen2.5 are recent additions to this series, |
| qwen-2.5-72b | 72b | Qianwen | featuring strong performance. |
| mistral-7b-v0.3 | 7b | Apache 2.0 | Instruction-tuned version of Mistral-7B v0.3 model (Jiang et al., 2023) from Mistral AI. |
| deepseek-v3 | - | DeepSeek License | DeepSeek V3 represents the latest models from DeepSeek AI, building on their V2 architecture (DeepSeek-AI et al., 2024). |
| mistral-nemo | 12b | Mistral AI Non-Prod. | Mistral Nemo is a 12B parameter model developed by Mistral AI in partnership with NVIDIA Blog. |
| gemini-2.0-flash | - | Proprietary | Gemini 2.0 Flash is part of Google's latest generation of capable multimodal models (Gemini et al., 2023). |
| claude-3.5-sonnet | - | Proprietary | Claude 3.5 and 3.7 Sonnet are advanced proprietary |
| claude-3.7-sonnet | - | Proprietary | models by Anthropic PBC (Claude, 2023). |
| gpt-4o | - | Proprietary | GPT-4o and GPT-4.1 are powerful proprietary models |
| gpt-4.1 | - | Proprietary | from OpenAI (Achiam et al., 2023). |

inferred preference is then compared to the ground-truth human label, allowing us to use the same accuracy metric as in our main pairwise experiments.

As shown in Table 12, the reference-guided **RefEval-point** method consistently outperforms the reference-free **Base-point** baseline for both the average of 11 open-source models and for the stronger frontier model group (GPT4o and GPT-4.1).

## C  ADDITIONAL ANALYSES AND ABLATIONS

### C.1  IMPACT OF REFERENCE QUALITY

**Evaluation Robustness.**   We investigated whether our proposed **RefEval** method relies heavily on "Oracle" quality references (e.g., GPT-4o). Table 14 shows the performance of Llama-3.1-70B and Qwen-2.5-7B as judges when guided by references from significantly smaller/weaker models (Mistral-Nemo-12B and Tulu-2-7B). The results demonstrate that while stronger references yield the highest accuracy, even references from 7B models provide a clear signal that improves over the reference-free baseline.

**Training Robustness.**   We further tested the robustness of our reference-guided self-improvement pipeline (§**??**) by replacing the DeepSeek-V3 references with those generated by **GPT-4o-mini**. Table 15 shows that even with weaker references, the **RefEval** pipeline enables effective self-improvement, outperforming the reference-free equivalent.

Table 14: Ablation of Reference Source Quality on Evaluation Accuracy (Avg over 4 datasets: Nat, Adv, MT, Ins).

| Judge Model | Reference Source | Method | Avg. Acc. |
|---|---|---|---|
| Llama-3.1-70B | GPT-4o (Oracle) | RefEval | 0.856 |
| | Nemo-12B | RefEval | 0.850 |
| | Tulu-2-7B | RefEval | 0.834 |
| | *None* | *Ref-Free* | *0.832* |
| Qwen-2.5-7B | GPT-4o (Oracle) | RefEval | 0.735 |
| | Nemo-12B | RefEval | 0.726 |
| | Tulu-2-7B | RefEval | 0.731 |
| | *None* | *Ref-Free* | *0.710* |

Table 15: Ablation of Reference Source Quality on Training (Llama-3-8B-Instruct).

| Method | AlpacaEval | Arena-Hard |
|---|---|---|
| Base (Llama-3-8B-Inst) | 25.0 | 27.1 |
| *Reference Source: DeepSeek-V3* | | |
| V3-Distill | 53.9 | 42.2 |
| V3-RefFree | 67.5 | 53.8 |
| **V3-RefEval** | **73.1** | **58.7** |
| *Reference Source: GPT-4o-mini* | | |
| 4o-mini-Distill | 28.7 | 40.7 |
| 4o-mini-RefFree | 42.6 | 41.7 |
| **4o-mini-RefEval** | **44.4** | **58.3** |

## C.2 INTER-JUDGE AGREEMENT

We analyzed the consistency of open-source judges by computing the average pairwise agreement between all 11 models. Table 16 shows that **RefEval** significantly increases agreement across all datasets compared to **Ref-Free**, suggesting that references reduce the variance in subjective judgment.

Table 16: Average Pairwise Inter-Judge Agreement (%) among 11 open-source models.

| Dataset Group | Ref-Free | RefEval | Difference |
|---|---|---|---|
| LLMBar-Natural | 80.98 | 85.31 | +4.33 |
| LLMBar-Adversarial | 75.59 | 77.61 | +2.02 |
| MTBench | 75.35 | 82.42 | +7.07 |
| Instrusum | 76.95 | 79.69 | +2.74 |
| HREF | 74.17 | 81.83 | +7.66 |
| **Average** | **76.61** | **81.37** | **+4.76** |

## C.3 MARGINAL GAINS OF MULTI-REFERENCE VOTING

We analyze the performance gain from increasing the number of references used in a voting ensemble (Figure 3). Table 17 shows the average accuracy of **RefEval** using voting ensembles of increasing size, drawn from a pool of diverse frontier models. While adding references consistently improves performance, the marginal gain diminishes, justifying our focus on single-reference efficiency in the main experiments.

## D DATASET DETAILS

This section provides further details on the datasets used for evaluating the LLM-as-a-Judge protocols described in §3.3. All datasets consist of instances with an input instruction and two candidate model

Table 17: Marginal gains from increasing reference count (Average across 11 judges).

| Configuration | # Refs | Avg. Accuracy | Gain |
|---|---|---|---|
| Single Best (Avg) | 1 | 81.4% | - |
| Pairwise Ensemble (Simulated) | 2 | 81.8% | +0.4% |
| Majority Vote | 3 | 82.2% | +0.4% |
| Majority Vote | 5 | 82.3% | +0.1% |

outputs, along with a human preference label indicating which output is superior or if they are tied. For our primary evaluation metric (accuracy), ties are typically excluded or handled according to the original benchmark's protocol if specified for pairwise win-rate calculations. Our reported accuracies are averaged over two evaluation passes, swapping the order of candidate outputs.

## D.1 CORE EVALUATION DATASETS

**LLMBar-Natural (Nat) and LLMBar-Adversarial (Adv)** (Zeng et al., 2024): These datasets were designed for meta-evaluating LLM evaluators on instruction following. **LLMBar-Natural** comprises 100 instances collected and filtered from existing human preference datasets, focusing on objective quality differences. **LLMBar-Adversarial** contains 319 instances where the dispreferred output is adversarially crafted to possess superficially appealing qualities (e.g., engaging tone, better formatting) that might mislead an LLM judge, despite deviating from the instruction. Both datasets feature high inter-annotator agreement (90% for Natural, 95% for Adversarial). We utilize the provided pairwise human preference labels.

**MTBench (MT)** (Zheng et al., 2024): This benchmark consists of 80 unique multi-turn conversation prompts spanning eight categories (e.g., writing, roleplay, math, coding). For each prompt, responses from various models are collected. The evaluation involves pairwise comparisons of these responses, judged by strong LLMs (typically GPT-4) based on human-defined criteria, simulating expert human judgments. We use the 200 expert-annotated pairwise comparisons (excluding ties) from the publicly released data, which have an 81% inter-annotator agreement rate.

**Instrusum** (Liu et al., 2024b): This dataset focuses on instruction-controllable summarization. Each instance includes a source document, a specific summarization instruction (e.g., varying length, style, or focus), and model-generated summaries. Human annotators provide pairwise preference labels for summaries based on adherence to the instruction and overall quality. We use the 411 instances from this dataset that have perfect inter-annotator agreement (100%) to ensure a high-quality, low-noise signal for our meta-evaluation. The instructions in Instrusum are notably longer and more complex on average compared to other datasets.

**HREF** (Lyu et al., 2024): The Human Response-Guided Evaluation of Instruction Following (HREF) benchmark provides human-written instructions and corresponding human-written reference responses. For our study, we selected a subset of the HREF human agreement set, which contains pairwise comparisons of model-generated outputs against these instructions, annotated by humans for preference. Specifically, we utilized instances from the following five task categories: 1) Classification (cls), 2) Closed QA (cqa), 3) Extraction (ext), 4) Generation (gen), 5) Rewriting (rew).

We excluded task categories such as summarization (to avoid overlap with **Instrusum**), brainstorming (brn), and open QA (oqa) to maintain dataset diversity and focus, or due to their more subjective nature which might introduce variance and noisiness.

We present the number of instances for each dataset in Table 18.

## D.2 CREATION OF HUMAN ORACLE REFERENCES

In Section 3.6, we investigate the impact of exceptionally high-quality human references, particularly for strong LLM-judges on challenging tasks, we created "Human Oracle" references for a subset of the LLMBar-Adversarial dataset (Zeng et al., 2024).

The creation process involved randomly selecting 23 instances where our standard GPT-4o judge (using its own generated reference) had previously made evaluation errors against the dataset's

Table 18: Number of instances for each evaluation dataset detailed in Appendix D.

| Dataset | No. of Instances |
|---|---|
| LLMBar-Natural (Nat) | 100 |
| LLMBar-Adversarial (Adv) | 319 |
| MTBench (MT) | 200 |
| Instrusum (Ins) | 411 |
| HREF - CLS | 56 |
| HREF - CQA | 73 |
| HREF - EXT | 64 |
| HREF - GEN | 70 |
| HREF - REW | 92 |
| Total | 1385 |

original human annotations. For these selected instances, an NLP expert (a co-author of the paper) revised the initial GPT-4o-generated references. Critically, this revision was performed *blindly*: the expert was provided only with the original instruction for each instance and did not see the candidate model outputs (Output A and Output B) or the original human preference label. The task was to create an optimal reference answer for the given instruction, focusing on correctness, completeness, and precise adherence to all instructional requirements. This methodology ensured the oracle references were developed independently of the specific outputs they would later be used to evaluate, providing a rigorous test of reference quality impact.

## E    TRAINING DETAILS

Here, we provide additional training details (§ 4.2). We use number of epochs of 2, a batch size of 128, a maximum learning rate of 5e-6, with a linear learning rate scheduler and 3% warmup steps. The maximum sequence length is 2048 tokens and the training instances that exceed this length are filtered out, resulting in 883K training instances.

## F    MODEL REGISTRY

Table 13 lists details on the models used in this research.

## G    DETAILS OF INSTRUCTION CLASSIFICATION FOR ALPACAEVAL AND ARENA-HARD

In §4.3, we classify the instructions in AlpacaEval and Arena-Hard into different categories to further compare the reference-based LLM-judges' performance against the baselines. Specifically, we use GPT-4o to classify the instructions into 4 categories: Coding&Math, Creative Tasks, Information Seeking, and Reasoning&Planning. The prompt template used is shown in Figure 8.

Figure 6 and Figure 7 demonstrate the instruction type distributions of AlpacaEval and Arena-Hard, respectively. They show that AlpacaEval contains more open-ended instructions, while Arena-Hard has a larger emphasis on coding and math reasoning.

## H    PROMPT TEMPLATES

This section provides the prompt templates used for our proposed methods and selected baselines discussed in the main paper. Placeholders like **{INSTRUCTION}**, **{OUTPUT_1}**, **{OUTPUT_2}**, and **{REFERENCE}** are filled at runtime.

Below we provide all the prompt templates used in this work.

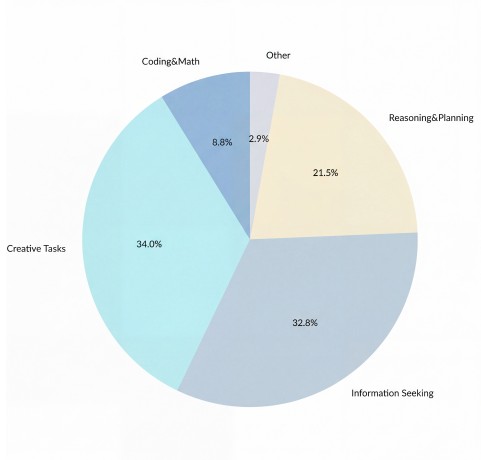

Figure 6: Distribution of different instruction types in AlpacaEval.

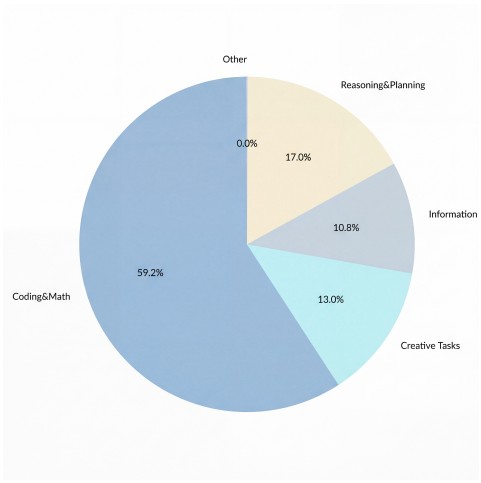

Figure 7: Distribution of different instruction types in Arena-Hard.

1. Ref-Free (Ours): Figure 9.
2. RefEval: Figure 10.
3. RefMatch: Figure 11.
4. HREF-Base (Lyu et al., 2024):Figure 12.
5. HREF-Ref (Lyu et al., 2024): Figure 13.
6. Multi-Ref Avg: Figure 14.
7. Multi-Ref Max: Figure 15.
8. RefMatch-Rules-CoT: Figure 16.
9. RefEval-Rules: Figure 17.
10. Metric-Ref: Figure 19.
11. Prepair: Figure 20 and Figure 21.
12. Self-Ref (Zeng et al., 2024): Figure 22.

---

**Instruction Classification**

**User Message:**
You are a helpful assistant that classifies prompts/instructions into one of these 4 categories:

1. Coding & Math
2. Information Seeking
3. Reasoning & Planning
4. Creative Tasks

Respond with ONLY ONE of these category NUMBERS (1, 2, 3, or 4), with no additional explanation or text.

---

Figure 8: Utility prompt – classification of instruction type introduced in §4.3.

---

**Ref-Free (Ours)**

**System Message:**
You are a helpful assistant that helps rate AI models' responses to instructions.

**User Message:**
Decide which output is better at following the instruction. The two outputs are generated by two different AI chatbots respectively.

Here are some aspects to consider:

1. Outputs should precisely follow the instruction. If an output contains unrelated information or does not complete each and all requirements in the instruction, it means that output does not precisely follow the instruction.

2. You should check for factual correctness and accuracy of outputs. If an output contains factual errors (especially with numbers), it should be considered lower quality.

3. Outputs should contain only a brief effective response without any verbose explanation, unless the instruction explicitly asks for an explanation.

4. The order in which the outputs are presented to you should NOT affect your judgment.

Select which output, "Output (a)" or "Output (b)", is better at following the instruction. Your answer should ONLY contain: "Output (a)" or "Output (b)".

# Instruction:
{INSTRUCTION}

# Output (a):
{OUTPUT_1}

# Output (b):
{OUTPUT_2}

# Which is the better, "Output (a)" or "Output (b)"? Your answer should ONLY contain either "Output (a)" or "Output (b)":

---

Figure 9: Prompt template for our **Ref-Free (Ours)** baseline prompting method introduced in §3.

---

**RefEval**

**System Message:**
You are a helpful assistant that helps rate AI models' responses to instructions.

**User Message:**
Decide which output is better at following the instruction. The two outputs are generated by two different AI chatbots respectively.

An effective and factually correct Reference Output is provided to aid your evaluation. This Reference Output demonstrates successful instruction-following.
Here are some aspects to consider:

1. Outputs should precisely follow the instruction. If an output contains unrelated information or does not complete each and all requirements in the instruction, it means that output does not precisely follow the instruction.

2. You should check for factual correctness and accuracy of outputs. If an output contains factual errors (especially with numbers), it should be considered lower quality. Compare the output against the Reference Output to verify if that output is factually correct.

3. Outputs should contain only a brief effective response without any verbose explanation, unless the instruction explicitly asks for an explanation.

4. Understand how the Reference Output properly delivers a helpful, accurate, and natural response, and then compare how closely an output matches this successful Reference Output.

5. Extraneous content in an output that goes beyond what is present in the Reference Output should be discouraged.

6. The order in which the outputs are presented to you should NOT affect your judgment.

Select which output, "Output (a)" or "Output (b)", is better at following the instruction. Your answer should ONLY contain: "Output (a)" or "Output (b)".

# Instruction:
{INSTRUCTION}

# Reference Output:
{REFERENCE}

# Output (a):
{OUTPUT_1}

# Output (b):
{OUTPUT_2}

# Which is the better, "Output (a)" or "Output (b)"? Your answer should ONLY contain either "Output (a)" or "Output (b)":

Figure 10: Prompt template for our **RefEval** prompting method introduced in §3.

---

**RefMatch**

**System Message:**
You are a helpful assistant tasked with comparing how similar two outputs are to a ground-truth Reference Output. Your goal is to determine which output demonstrates closer similarity to the reference.

**User Message:**
You will be given Output (a) and Output (b) for the Instruction, and a ground-truth Reference Output. Rules for similarity comparison:

1. The Instruction determines what to match for - extraneous information or incorrect number of elements means no match, even if there are word overlaps

2. Surface-level similarities (word matches, format) are not considered matches if they don't satisfy the Instruction requirements

3. First understand how the ground-truth Reference Output properly follows the Instruction to see what a successful answer looks like, then compare how closely Output (a) and Output (b) match this proper instruction-following pattern

4. Extraneous content in an output that goes beyond what is present in the Reference Output should be discouraged

Compare how each output relates to the ground-truth Reference Output. Before comparison, identify which aspects of the ground-truth Reference Output are essential to match given the context of the Instruction.

Then determine which output demonstrates closer similarity to the ground-truth Reference Output. You should answer using ONLY "Output (a)" or "Output (b)". Do NOT output any other words.

# Ground-truth Reference Output:
{REFERENCE}

# Instruction:
{INSTRUCTION}

# Output (a):
{OUTPUT_1}

# Output (b):
{OUTPUT_2}

# Which is more similar to the Reference Output, Output (a) or Output (b)? Your response should ONLY be either "Output (a)" or "Output (b)" verbatim:

Figure 11: Prompt template for the **RefMatch** prompting method introduced in §3.

**HREF-Base**(Lyu et al., 2024)

**System Message:**
You are a helptul assistant that helps us rate an Al model's responses to instructions.

**User Message:**
Decide which response from the Al system following the instruction is better, considering the following questions:

1. Does the response precisely follow the instruction? For example, a response that includes unrelated information or does not fulfill the task is not precisely following the instruction.

2. Is the response helpful? For example, if the instruction asks for a recipe for healthy food, and the response is a useful recipe, then you can consider it helpful.

3. Is the response language natural? For example, Al responses are often verbose or repetitive, which is not natural.

4. Is the response factual/accurate? AI responses often make up new information. For example, if the response claims that Donald Trump is the current U.S. president, then you should consider it inaccurate.

5. Based on your aesthetics, which one do you prefer? For example, you might prefer one poem over another poem.

Select the response A or B that you prefer. Your answer should ONLY contain: A or B.

Now is the real task, just select among: A or B.

# Task:
## Instruction:
{INSTRUCTION}

## Response A:
{OUTPUT_1}

## Response B:
{OUTPUT_2}

## Which is the best, "A" or "B"? Your answer should ONLY contain either "A" or "B":

Figure 12: Prompt template for the **HREF-Base** prompting method from Lyu et al. (2024).

**HREF-Ref** (Lyu et al., 2024)

**System Message:**
You are a helpful assistant that helps us rate an AI model's responses to instructions.

**User Message:**
Decide which response from the AI system following the instruction is better, considering the following questions:

1. Does the response precisely follow the instruction? For example, a response that includes unrelated information or does not fulfill the task is not precisely following the instruction. Compare each response with the provided human response to decide if a response faithfully follows the instruction, especially when the instruction asks for expected word count or format.

2. Is the response helpful? For example, if the instruction asks for a recipe for healthy food, and the response is a useful recipe, then you can consider it helpful.

3. Is the response language natural? For example, AI responses are often verbose or repetitive, which is not natural. Compare with the provided human response to decide whether a response is natural.

4. Is the response factual/accurate? AI responses often make up new information. For example, if the response claims that Donald Trump is the current U.S. president, then you should consider it inaccurate. Compare with the provided human response to verify whether a response is factual and accurate, especially with numbers.

5. Based on your aesthetics, which one do you prefer? For example, you might prefer one poem over another poem.

Select the response A or B that you prefer. Your answer should ONLY contain: A or B.

Now is the real task, just select among: A or B.

# Task:
## Instruction:
{INSTRUCTION}

## Response A:
{OUTPUT_1}

## Response B:
{OUTPUT_2}

## Human Response:
{REFERENCE}

## Which is the best, "A" or "B"? Your answer should ONLY contain either "A" or "B":

Figure 13: Prompt template for the **HREF-Ref** prompting method from Lyu et al. (2024).

---

**Multi-Ref Avg**

**System Message:**
You are an expert AI assistant tasked with identifying which of two outputs more closely matches multiple Reference Outputs for a given Instruction.

**User Message:**
You will be given an Instruction, Output (a), Output (b), and three Reference Outputs.
Determine which of Output (a) or Output (b) has a higher degree of similarity to the set of Reference Outputs, while considering the context of the Instruction.

# Instruction:
{INSTRUCTION}

# Reference Output 1:
{REFERENCE_1}

# Reference Output 2:
{REFERENCE_2}

# Reference Output 3:
{REFERENCE_3}

# Output (a):
{OUTPUT_1}

# Output (b):
{OUTPUT_2}

# Decision (You should carry out a concise reasoning. Conclude your reasoning with either "Therefore, Output (a) is overall more similar to the Reference Outputs." or "Therefore, Output (b) is overall more similar to the Reference Outputs." VERBATIM. Always state which is more similar at the end. In your explanation, always use "Output (a)" or "Output (b)" to refer to the two outputs.):

---

Figure 14: Prompt template for the **Multi-Ref Avg** prompting method. Details in §A.2

---

**Multi-Ref MAX**

**System Message:**
You are an expert AI assistant tasked with selecting the output that best matches any of the provided Reference Outputs.

**User Message:**
You will be given candidate Output (a) and candidate Output (b) for the Instruction, and three Reference Outputs.
Compare candidate Output (a) and candidate Output (b) to each of the three Reference Outputs individually. Identify if there's a standout match between any candidate output (a or b) and any single Reference Output. Select the candidate output that matches best with ANY ONE of the Reference Outputs while considering the context of the Instruction.
# Instruction:
{INSTRUCTION}
# Output (a):
{OUTPUT_1}
# Output (b):
{OUTPUT_2}
# Reference Output 1:
{REFERENCE_1}
# Reference Output 2:
{REFERENCE_2}
# Reference Output 3:
{REFERENCE_3}
# Decision (You should carry out a concise reasoning. Conclude your reasoning with either "Therefore, Output (a) has a best match." or "Therefore, Output (b) has a best match." verbatim. Always state which has a best match AT THE END. In your explanation, always use "Output (a)" or "Output (b)" to refer to the two outputs.):

---

Figure 15: Prompt template for the **Multi-Ref MAX** prompting method. Details in §A.2.

---

**RefMatch-Rules-CoT**

**System Message:**
You are an expert AI assistant tasked with comparing how similar two outputs are to a Reference Output. Your goal is to determine which output demonstrates closer similarity to the reference.

**User Message:**
You will be given Output (a) and Output (b) for the Instruction, and a Reference Output.

Rules for similarity comparison:

1. The Instruction determines what to match for - extraneous information or incorrect number of elements means no match, even if there are word overlaps

2. Surface-level similarities (word matches, format) are not considered matches if they don't satisfy the Instruction requirements

3. First understand how the Reference Output properly follows the Instruction, then compare similarity based on this proper instruction-following

Compare how each output relates to the Reference Output.

Before comparison, identify which aspects of the Reference Output are essential to match given the context of the Instruction. Then determine which output demonstrates closer similarity to the Reference Output.

# Instruction:
{INSTRUCTION}

# Output (a):
{OUTPUT_1}

# Output (b):
{OUTPUT_2}

# Reference Output:
REFERENCE

# Decision (You should carry out a brief reasoning. Conclude your reasoning with either "Therefore, Output (a) shows closer similarity to the Reference Output." or "Therefore, Output (b) shows closer similarity to the Reference Output." VERBATIM. Always state which shows closer similarity at the end. In your explanation, always use "Output (a)" or "Output (b)" to refer to the two outputs.):

Figure 16: Prompt template for the **RefMatch-Rules-CoT** prompting method variant. §A.2

---

**RefEval-Rules**

**System Message:**
You are a helpful assistant in evaluating the quality of the outputs for a given instruction. Your goal is to select the best output for the given instruction.

**User Message:**
Select the Output (a) or Output (b) that is better for the given instruction. The two outputs are generated by two different AI chatbots respectively.

A ground-truth Reference Output is provided to aid your evaluation. This Reference Output demonstrates successful instruction-following and can help inform your judgment.

Here are some rules of the evaluation:

1. You should prioritize evaluating whether the output honestly/precisely/closely executes the instruction, then consider its helpfulness, accuracy, level of detail, harmlessness, etc.

2. Outputs should NOT contain more/less than what the instruction asks for, as such outputs do NOT precisely execute the instruction.

3. You should avoid any potential bias and your judgment should be as objective as possible. For example, the order in which the outputs were presented should NOT affect your judgment, as Output (a) and Output (b) are \*\*equally likely\*\* to be the better.

4. Use the ground-truth Reference Output to understand what a successful answer looks like. Evaluate whether Output (a) or Output (b) achieves similar effectiveness as the Reference Output in addressing the instruction's requirements.

Do NOT provide any explanation for your choice. Do NOT say both / neither are good.

You should answer using ONLY "Output (a)" or "Output (b)". Do NOT output any other words.

# Instruction:
{INSTRUCTION}

# Output (a):
{OUTPUT_1}

# Output (b):
{OUTPUT_2}

# Ground-truth Reference Output:
{REFERENCE}

# Which is better, Output (a) or Output (b)? Your response should be either "Output (a)" or "Output (b)":

---

Figure 17: Prompt template for the **RefEval-Rules** prompting method variant. §A.2

---

**CoT**

**[System Message]**
You are a helpful assistant in evaluating the quality of the outputs for a given instruction. Your goal is to select the best output for the given instruction.

**[User Message]**
After giving a brief explanation, select the Output (a) or Output (b) that is better for the given instruction. The two outputs are generated by two different AI chatbots respectively.

Here are some rules of the evaluation:
(1) You should prioritize evaluating whether the output honestly/precisely/closely executes the instruction, then consider its helpfulness, accuracy, level of detail, harmlessness, etc.
(2) Outputs should NOT contain more/less than what the instruction asks for, as such outputs do NOT precisely execute the instruction.
(3) You should avoid any potential bias and your judgment should be as objective as possible. For example, the order in which the outputs were presented should NOT affect your judgment, as Output (a) and Output (b) are equally likely to be the better.

You should first provide a brief explanation of your evaluation, and then always end your response with either "Therefore, Output (a) is better." or "Therefore, Output (b) is better." verbatim.
Do NOT say both / neither are good.
Do NOT output any other words.
Do NOT say "Output (a) is better" or "Output (b) is better" at the beginning. You should do reasoning and thinking before claiming which is better.

# Instruction:
{INSTRUCTION}

# Output (a):
{OUTPUT_1}

# Output (b):
{OUTPUT_2}

# Decision (Give a brief explanation of your evaluation followed by either "Therefore, Output (a) is better." or "Therefore, Output (b) is better." verbatim. Always claim which is better at the end. In your explanation, you should always use "Output (a)" or "Output (b)" to refer to the two outputs respectively.):

Figure 18: Prompt for **cot** prompting method described in §3

---

**Metric + Reference**

**[System Message]**
You are a helpful assistant in evaluating the quality of the outputs for a given instruction. Your goal is to select the best output for the given instruction.

**[User Message]**
Select the Output (a) or Output (b) that is better for the given instruction. The two outputs are generated by two different AI chatbots respectively.

Here are some rules of the evaluation:
(1) You should prioritize evaluating whether the output honestly/precisely/closely executes the instruction, then consider its helpfulness, accuracy, level of detail, harmlessness, etc.
(2) Outputs should NOT contain more/less than what the instruction asks for, as such outputs do NOT precisely execute the instruction.
(3) You should avoid any potential bias and your judgment should be as objective as possible. For example, the order in which the outputs were presented should NOT affect your judgment, as Output (a) and Output (b) are equally likely to be the better.

Do NOT provide any explanation for your choice.
Do NOT say both / neither are good.
You should answer using ONLY "Output (a)" or "Output (b)". Do NOT output any other words.

# Instruction:
{INSTRUCTION}

# Output (a):
{OUTPUT_1}

# Output (b):
{OUTPUT_2}

# Questions about Outputs:
Here are at most three questions about the outputs, which are presented from most important to least important. You can do the evaluation based on thinking about all the questions.
{QUESTIONS}

# A reference output generated by a strong AI assistant:
{REFERENCE}

# Which is better, Output (a) or Output (b)? Your response should be either "Output (a)" or "Output (b)":

Figure 19: Prompt for **metric + reference** prompting method described in §3.

---

**Prepair (pointwise analysis)**

**[System Message]**
You are a helpful assistant in evaluating the quality of the outputs for a given instruction. Your goal is to evaluate the quality of output for the given instruction.

**[User Message]**
Giving a brief explanation to evaluate the quality of the response to the given instruction. The output is generated by an AI chatbot.

Here are some rules of the evaluation:
(1) You should prioritize evaluating whether the output honestly/precisely/closely executes the instruction, then consider its helpfulness, accuracy, level of detail, harmlessness, etc.
(2) The model outputs should NOT contain more/less than what the instruction asks for, as such outputs do NOT precisely execute the instruction.
(3) You should avoid any potential bias and your judgment should be as objective as possible.

You should provide a brief explanation of your evaluation.
Your explanation should identify critical drawbacks in model outputs that do not meet the above evaluation rules.

# Instruction:
{INSTRUCTION}

# Output:
{OUTPUT}

# Provide your explanation:

---

Figure 20: Prompt for **prepair** prompting method described in §3. This is the prompt for pointwise analysis (the first stage) within the method.

---

**Prepair (pairwise evaluation)**

**[System Message]**
You are a helpful assistant in evaluating the quality of the outputs for a given instruction. Your goal is to select the best output for the given instruction.

**[User Message]**
After giving a brief explanation, select the Output (a) or Output (b) that is better for the given instruction. The two outputs are generated by two different AI chatbots respectively.

Here are some rules of the evaluation:
(1) You should prioritize evaluating whether the output honestly/precisely/closely executes the instruction, then consider its helpfulness, accuracy, level of detail, harmlessness, etc.
(2) Outputs should NOT contain more/less than what the instruction asks for, as such outputs do NOT precisely execute the instruction.
(3) You should avoid any potential bias and your judgment should be as objective as possible. For example, the order in which the outputs were presented should NOT affect your judgment, as Output (a) and Output (b) are **equally likely** to be the better.

You should first provide a brief explanation of your evaluation, and then always end your response with either "Therefore, Output (a) is better." or "Therefore, Output (b) is better." verbatim.
Do NOT say both / neither are good.
Do NOT output any other words.
Do NOT say "Output (a) is better" or "Output (b) is better" at the beginning.

You should do reasoning and thinking **before** claiming which is better. Your explanation should identify critical drawbacks in model outputs that do not meet the above evaluation rules.

# Instruction:
{INSTRUCTION}

# Output (a):
{OUTPUT_1}

# Output (b):
{OUTPUT_2}

# Here's the analysis for each output you wrote earlier:
{PER OUTPUT ANALYSES}

# Your Response (Provide your evaluation and reasoning, followed by either "Therefore, Output (a) is better." or "Therefore, Output (b) is better." verbatim):

---

Figure 21: Prompt for **prepair** prompting method described in §3. This is the pairwise evaluation stage (the second stage) within the method.

---

**Self-Ref**

**[System Message]**
You are a helpful assistant in evaluating the quality of the outputs for a given instruction. Your goal is to select the best output for the given instruction.

**[User Message]**
Select the Output (a) or Output (b) that is better for the given instruction. The two outputs are generated by two different AI chatbots respectively.

Here are some rules of the evaluation:
(1) You should prioritize evaluating whether the output honestly/precisely/closely executes the instruction, then consider its helpfulness, accuracy, level of detail, harmlessness, etc.
(2) Outputs should NOT contain more/less than what the instruction asks for, as such outputs do NOT precisely execute the instruction.
(3) You should avoid any potential bias and your judgment should be as objective as possible. For example, the order in which the outputs were presented should NOT affect your judgment, as Output (a) and Output (b) are equally likely to be the better.

Do NOT provide any explanation for your choice.
Do NOT say both / neither are good.
You should answer using ONLY "Output (a)" or "Output (b)". Do NOT output any other words.

# Instruction:
{INSTRUCTION}

# Output (a):
{OUTPUT_1}

# Output (b):
{OUTPUT_2}

# A reference output generated by a strong AI assistant:
{REFERENCE}

# Which is better, Output (a) or Output (b)? Your response should be either "Output (a)" or "Output (b)":

Figure 22: Prompt for **Self-Ref** prompting method described in §3.

---

**Base-point**

**System Message:**
You are a helpful assistant in evaluating the quality of the outputs for a given instruction. Your goal is to score the output for the given instruction.

**User Message:**
Score the Output on a Likert scale from 1 to 5 for the given instruction, where a score of one means "poor quality" and score of five means "perfect quality". The output is generated by an AI chatbot.

Here are some rules of the evaluation:

1. You should prioritize evaluating whether the output honestly/precisely/closely executes the instruction, then consider its helpfulness, accuracy, level of detail, harmlessness, etc.

2. Outputs should NOT contain more/less than what the instruction asks for, as such outputs do NOT precisely execute the instruction.

3. You should avoid any potential bias and your judgment should be as objective as possible.

Do NOT provide any explanation for your choice. You should answer 1, 2, 3, 4, or 5 only. Do NOT output any other words.

# Instruction:
{INSTRUCTION}

# Output:
{OUTPUT}

# What is your rating for the Output?

---

Figure 23: Prompt for **Base-point** prompting method described in §B.

---

**RefEval-point**

**System Message:**
You are a helpful assistant that helps rate an AI model's response to an instruction.

**User Message:**
Score the given Output on a Likert scale from 1 to 5, where a score of one means "very poor quality" and a score of five means "perfect quality".

An effective and factually correct Reference Output is provided to aid your evaluation. This Reference Output demonstrates successful instruction-following.

Here are some aspects to consider when scoring:

1. The Output should precisely follow the instruction. If the Output contains unrelated information or does not complete each and all requirements in the instruction, it should be scored lower.

2. You must check for factual correctness and accuracy. If the Output contains factual errors, it should be considered lower quality. Compare the Output against the Reference Output to verify factual correctness.

3. The Output should contain only a brief effective response without any verbose explanation, unless the instruction explicitly asks for one.

4. Understand how the Reference Output properly delivers a helpful, accurate, and natural response, and then evaluate how closely the given Output matches this successful Reference Output.

5. Extraneous content in the Output that goes beyond what is present in the Reference Output should be discouraged and result in a lower score.

You should answer with a single digit from 1 to 5 only. Do NOT provide any explanation for your choice.

# Instruction:
{INSTRUCTION}

# Reference Output:
{REFERENCE}

# Output to Score:
{OUTPUT}

# What is your rating for the Output?

---

Figure 24: Prompt for **RefEval-point** prompting method described in §B.

