# OpenReview forum: "References Improve LLM Alignment in Non-Verifiable Domains"
_ICLR.cc/2026/Conference — ICLR 2026 Poster_

### Official Review · Reviewer_P3EF · 2025-10-27

**Soundness:** 2
**Presentation:** 2
**Contribution:** 2
**Rating:** 2
**Confidence:** 4

**Summary:**

This paper studies how high-quality reference outputs can be used to improve LLM alignment. The authors first design prompting strategies that allow LLM-based evaluators to compare candidate answers with a reference, showing consistent accuracy gains across 11 open-source models on 5 datasets. They then extend the idea to training, where the model generates multiple outputs, ranks them with a reference-guided judge, and performs preference optimization (DPO). Experiments on AlpacaEval and Arena-Hard show that reference-guided self-improvement outperforms reference-free methods and reaches performance comparable to reward-model-based approaches, without requiring additional human feedback.

**Strengths:**

**Strength 1**: The study evaluates 11 open-source LLMs across five human-annotated datasets (LLMBar-Natural/Adversarial, MTBench, InstruSum, HREF), covering diverse architectures and capability tiers. This broad and systematic evaluation strengthens result credibility and supports reproducibility.

**Strength 2**: Reference-guided approaches such as RefEval and RefMatch consistently outperform both reference-free methods and prior reference-based baselines. Evidence from Table 1, Table 2, and Table 4 shows notable gains in both judgment accuracy and downstream alignment tuning, highlighting the practical effectiveness of references.

**Strength 3**: The analysis reveals stronger relative gains for smaller models and demonstrates that higher-quality references (e.g., human-edited) further improve judge performance. Results in Table 3 and Figure 4 provide helpful insights into when reference guidance is most beneficial.

**Weaknesses:**

**Weakness 1**: While the paper offers a more systematic and larger-scale study, the core concept of reference-guided judging remains closely related to prior work. In addition, recent studies like Self-Rationalization Improves LLM as a Fine-Grained Judge[1] similarly explore LLM-as-a-Judge and preference-based self-improvement. A clearer discussion of methodological differences and contribution boundaries would help clarify the incremental novelty introduced by leveraging external references.

**Weakness 2**: The pipeline uses frontier LLMs to generate references and also as strong evaluators in benchmarks like AlpacaEval and Arena-Hard. This may cause circular style or preference bias, making it difficult to confirm whether improvements reflect better human alignment or closer imitation of the reference model family.

**Weakness 3**: The paper does not provide clear criteria or measurements for selecting or validating high-quality references. This makes it difficult to assess generalization, especially when references may vary in coverage or factual reliability across datasets.

[1]Trivedi, Prapti, et al. "Self-rationalization improves llm as a fine-grained judge." arXiv preprint arXiv:2410.05495 (2024).

**Questions:**

Q1: What criteria are used to judge that a reference output is "high quality," and how sensitive is the method to different levels of reference quality?

Q2: Since GPT-4o is used to generate reference outputs and the evaluation benchmarks (e.g., AlpacaEval, Arena-Hard) are also judged by GPT-4 family models, how can we be sure that the improvements indicate stronger human preference alignment rather than closer imitation of GPT-4o-style responses? Incorporating human evaluation or cross-family judging would better support the alignment claims.

Q3: Can the proposed method work when references are unavailable or noisy in new domains, and what are the expected failure conditions when references deviate from correct or complete content?

---

> ### Author Response · Authors · 2025-11-26
> **Response to Reviewer P3EF [Part 1]**
>
> Thank you for your detailed review. We appreciate you noting our “notable gains in both judgment accuracy and downstream alignment.” We believe there are some misunderstandings regarding our experimental setup, which we clarify below.
>
> > “While the paper offers a more systematic and larger-scale study, the core concept of reference-guided judging remains closely related to prior work. In addition, recent studies like Self-Rationalization Improves LLM as a Fine-Grained Judge[1] similarly explore LLM-as-a-Judge and preference-based self-improvement. A clearer discussion of methodological differences and contribution boundaries would help clarify the incremental novelty introduced by leveraging external references. [1]Trivedi, Prapti, et al. "Self-rationalization improves llm as a fine-grained judge." arXiv preprint arXiv:2410.05495 (2024).”
>
> 1. Please refer to our general response regarding our unique contributions relative to prior reference-guided evaluation work. To the best of our knowledge, we are the first to **use reference-guided evaluation to enable LLM self-improvement in non-verifiable domains with high-quality references**, effectively bridging the gap between RLVR and training methods for non-verifiable tasks.
>
> 2. Regarding Trivedi et al., their method focuses on **internal self-rationalization** (via Chain-of-Thought). In contrast, our approach relies on **external grounding** through high-quality references. These address different limitations: internal reasoning vs. access to reliable knowledge or ground-truth signals. Furthermore, the self-improvement in Trivedi et al. applies to the **judge model**, whereas our primary contribution is enabling **self-improvement of the policy model** using reference-guided judges, an aspect not covered in their work.
>
> 3. We note that our methods also *outperform* Trivedi et al. under comparable settings. Specifically, with Llama-3.1-8B-Instruct as the base LLM, their fine-tuned judge achieves only 0.56 on the Chat-Hard subset of rewardbench, which consists mostly of data instances in LLMBar-Hard. In contrast, our reference-free baseline already achieves 0.629 on LLMBar-Hard, and our reference-aided method reaches 0.745. This suggests that the evaluation prompt used in Trivedi et al. is likely not well optimized.
>
> ---
> > "The pipeline uses frontier LLMs to generate references and also as strong evaluators in benchmarks like AlpacaEval and Arena-Hard. This may cause circular style or preference bias, making it difficult to confirm whether improvements reflect better human alignment or closer imitation of the reference model family." “Since GPT-4o is used to generate reference outputs and the evaluation benchmarks (e.g., AlpacaEval, Arena-Hard) are also judged by GPT-4 family models, how can we be sure that the improvements indicate stronger human preference alignment rather than closer imitation of GPT-4o-style responses? Incorporating human evaluation or cross-family judging would better support the alignment claims.”
>
> We believe there is a misunderstanding here. We note that the reference outputs used in training are **not** generated by GPT-4o, but by **DeepSeek-V3** (Line 394). The trained models are then evaluated on AlpacaEval and Arena-Hard using **GPT-4o**. Since these two models are from different model families, the risk of circularity is relatively low. Thank you for raising this concern.

---

> ### Author Response · Authors · 2025-11-26
> **Response to Reviewer P3EF [Part 2]**
>
> > "The paper does not provide clear criteria or measurements for selecting or validating high-quality references. This makes it difficult to assess generalization, especially when references may vary in coverage or factual reliability across datasets." “What criteria are used to judge that a reference output is ‘high quality,’ and how sensitive is the method to different levels of reference quality?”
>
> We appreciate the comment.
>
> 1. In our experimental settings, we rely on frontier models (DeepSeek-V3) to generate the references for training smaller LLMs (Llama-3-8B, Qwen2.5-7B). Since DeepSeek-V3 performs much better than these smaller models, we assume the references are of high quality in this setting.
>
> 2. The evaluation results on AlpacaEval and Arena-Hard, which use GPT-4o as the evaluator, also confirm the high quality of the reference outputs generated by DeepSeek-V3 (Table 5), as it achieves 84.8 and 94.9, respectively, meaning it outperforms the base model GPT-4 by a large margin.
>
> 3. As discussed in our general response, our original manuscript already conducted a study (Figure 3) showing that references generated by LLMs from different model families (e.g., GPT-4o, Claude-3.5, Gemini-2.0) can substantially improve the LLM-judge’s performance, indicating that our evaluation method generalizes to references with different characteristics. Moreover, we provided additional results in our general response showing that our method also benefits from references generated by less capable LLMs, although with smaller improvements as expected.
>
> 4. We agree that clear criteria or measurements for selecting or validating high-quality references could be beneficial for more rigorous evaluations. However, we believe this would require dedicated work on data quality, which is beyond the scope of our paper. We note that data quality is not unique to our work but is a common topic in post-training, especially for methods that use SFT.
>
> ---
> > “Can the proposed method work when references are unavailable or noisy in new domains, and what are the expected failure conditions when references deviate from correct or complete content?”
>
> Thank you for the questions!
>
> 1. We note that the reference quality and availability are a fundamental challenge for all supervised learning, not specific to our method. If a domain is so difficult that no references (human or model) exist, then even standard SFT or RLVR cannot be performed. Our contribution is to show that where SFT is possible (i.e., references exist), our method extracts more value from those references than SFT alone. Our method is analogous to RLVR in this sense – RLVR for domains such as math reasoning requires ground-truth answers, while our method requires high-quality references.
>
> 2. We have conducted additional ablations with weaker references (please refer to our general response). The results show that LLMs-as-Judges can also benefit from reference outputs generated by less capable LLMs. However, the improvement is less significant compared to high-quality references as expected.

---

> > ### Comment · Reviewer_P3EF · 2025-11-28
> >
> > Thank you for the clarifications. The responses clear up several of my earlier concerns. These additions make the empirical side of the paper stronger. Some concerns still exist. The conceptual novelty feels incremental, and the method depends a lot on having access to strong models for high-quality references. It would also be helpful to discuss how well the approach works for more open-ended or less structured tasks that can't be verified.

---

> > > ### Author Response · Authors · 2025-11-28
> > > **Reply to Reviewer P3EF**
> > >
> > > Thank you for your reply. We are encoruaged to see that our response has cleared up some of your concerns.
> > >
> > > > "The conceptual novelty feels incremental"
> > >
> > > Please refer to our general response, where we have provided a detailed discussion about the novelty and the contribution of our work.
> > >
> > > ---
> > > > "The method depends a lot on having access to strong models for high-quality references."
> > >
> > > Our general response also discussed the impact of reference quality on our method. We would like to provide further clarification here. Our method does not depend on access to strong models per se, but on having **high-quality references**. In our empirical experiments, we use strong models to generate references for smaller models, which allows for a controlled and scalable experimentation setup. Since the core idea of our method is to leverage references, this reliance on high-quality references is inherent by design.
> > >
> > > However, as noted in our general response, this assumption is **not unique to our method**. Standard SFT and RLVR also rely on high-quality reference outputs: in SFT, models are fine-tuned on reference outputs, and in RLVR, reference outputs effectively define the reward signal. Therefore, dependence on reference quality is a standard design choice shared across post-training methods, rather than a specific limitation of our approach.
> > >
> > > ---
> > > > "It would also be helpful to discuss how well the approach works for more open-ended or less structured tasks that can't be verified."
> > >
> > > Thank you for the suggestion. We note that the training dataset we use, UltraFeedBack, already contains data points for diverse, open-ended, and unstructured tasks. In Section 4.3, we have provided a study on the improvement of our method across different types of tasks. The results (Figure 4) show that our method improves over the reference-free baseline across all evaluated categories (Coding and Math, Creative Tasks, Information Seeking, Reasoning and Planning).

---

### Official Review · Reviewer_r7ga · 2025-10-28

**Soundness:** 4
**Presentation:** 4
**Contribution:** 4
**Rating:** 10
**Confidence:** 3

**Summary:**

The paper studies using references from frontier LLMs for LLM alignment. Specifically in domains where verifiable rewards are not available, the quality of less capable LLM evaluators can be much improved by guiding those with references from frontier LLMs.

**Strengths:**

1. The proposed approach is a step towards something similar to RLVR for non-verifiable domains, which makes this work interesting to the broad community.
2. Comparisons with strong baselines from literature are provided. In addition the authors design a strong reference-free baseline that is directly comparable to their referenced based approach in terms of prompt quality.
3. Experiments with sources from different frontier LLMs are conducted so that the results are not specific to a particular frontier LLM. This validates the the generalization of the proposed approach.
4. Finally, the paper shows results for actual preference alignment training using the reference guided evaluator. The results conclusively show that the proposed approach is superior.

**Weaknesses:**

1. The baselines in table 4 are not as strong. What I mean is that some of the baselines for table 4 should probably have been based on the baselines in table 1. If we are to choose a evaluator approach based on results from table 1, we would like to know whether the results in table 1 can serve as a robust tool for choosing a llm evaluator.

**Questions:**

In section 4.3 you wrote

 (3) V3-Distill is the SFT model
distilled from DeepSeek v3 references. The following models are finetuned from V3-Distill: (4) ROUGE,
which uses ROUGE scores as the reward5
; (5) BERTScore, which uses
the BERTScore metric (Zhang et al.,
2020; Zhao et al., 2025); (6) ArmoRM, which uses the ArmoRM reward model;

What does it mean to finetune ROUGE, BERTScore etc to be finetuned from V3-Distill?

---

> ### Author Response · Authors · 2025-11-26
> **Response to Reviewer r7ga**
>
> We are grateful for your strong support and for highlighting our work as a "step towards RLVR for non-verifiable domains."
>
> > “The baselines in table 4 are not as strong. What I mean is that some of the baselines for table 4 should probably have been based on the baselines in table 1. If we are to choose a evaluator approach based on results from table 1, we would like to know whether the results in table 1 can serve as a robust tool for choosing a llm evaluator.”
>
> We clarify our selection logic: In Table 4 (Training), the RefFree baseline uses the “Ref-Free (Ours)” prompting strategy, which is the strongest reference-free prompt evaluated in Table 2. In Table 2 (Evaluation), “Ref-Free (Ours)” achieves 73.7%, higher than both “LLMBar-Base” (72.3%) and “CoT” (71.2%). Moreover, the RefFree baseline and the RefEval method in Table 4 share similar rules and structures in their prompts, with the only difference being the rules for using references. This ensures a controlled comparison that isolates the impact of reference outputs on the LLM-judge used in preference optimization.
>
> ---
> > “In section 4.3 you wrote (3) V3-Distill is the SFT model distilled from DeepSeek v3 references. The following models are finetuned from V3-Distill: (4) ROUGE, which uses ROUGE scores as the reward5 ; (5) BERTScore, which uses the BERTScore metric (Zhang et al., 2020; Zhao et al., 2025); (6) ArmoRM, which uses the ArmoRM reward model; What does it mean to finetune ROUGE, BERTScore etc to be finetuned from V3-Distill?”
>
> Thank you for catching this ambiguity in our wording. We do not fine-tune ROUGE, BERTScore, or ArmoRM themselves. What we mean is that, starting from V3-Distill, we perform DPO training where the preference pairs are constructed and scored using ROUGE, BERTScore, or the ArmoRM reward model, respectively. We will revise Section 4.3 to make this clear and to emphasize that these experiments compare our reference-guided LLM-judge with traditional metric-based and reward-model-based supervision.

---

### Official Review · Reviewer_2bbc · 2025-11-01

**Soundness:** 3
**Presentation:** 3
**Contribution:** 2
**Rating:** 4
**Confidence:** 3

**Summary:**

The paper asks whether giving judges and students a good reference answer helps align smaller LLMs. It introduces three prompt styles for judging—Ref-Free (no reference), RefEval (compare to a reference), and RefMatch (pick the candidate closest to the reference)—and then uses references twice: (i) SFT to distill style/quality from strong references, and (ii) DPO where a reference-guided judge labels on-policy pairs for preference training. Across roughly 5 datasets and 11 judges, reference-guided prompts improve judge reliability, and the SFT→DPO pipeline lifts 7–8B models on common leaderboards (e.g., AlpacaEval, Arena-Hard). Overall, it’s a practical recipe: simple prompts, references as grounding, and a two-stage training loop that competes with a reward-model baseline without extra human feedback.

**Strengths:**

The paper asks a clear, practical question—can we ground LLM judges and training with strong references—and answers it with simple, reproducible tooling (Ref-Free, RefEval, RefMatch) and a clean SFT→DPO pipeline. The experimental setup is broad (many judges, several datasets) and the gains are consistent: reference-guided judging improves agreement/utility, and reference-distilled SFT followed by preference optimization moves mid-size models meaningfully on common leaderboards. I also like that prompts and templates are easy to reuse, and that multi-reference variants are explored rather than just a single “gold” answer.

**Weaknesses:**

Methodologically, the novelty feels incremental: reference-guided evaluation and reference-distilled training have both appeared before, and the paper mostly scales and systematizes them rather than introducing a new objective or learning signal. The comparisons also underplay strong preference-optimization baselines (e.g., SimPO, ORPO, KTO) and fine-grained supervision relevant to credit assignment (token/segment-level DPO; span-supervised MT like TWA).

Robustness is not fully stress-tested: results rely on high-quality references, but we don’t see sensitivity to noisy, weaker, or stylistically different references, nor clear evidence against style coupling to the reference generator. Finally, statistical reporting is light (few CIs/permutation tests), and most end metrics are still LLM-judge based; more targeted human spot-checks would help address circularity.

Missing citation and baseline:
RevisEval: Improving LLM-as-a-Judge via Response-Adapted References — Qiyuan Zhang, ICLR 2025.

**Questions:**

Reference robustness: How do results change if you paraphrase, shorten, or inject minor errors into references, or use weaker LLMs as references? Any curve of judge accuracy vs. reference quality?

Baseline coverage: Can you add at least one strong SimPO/ORPO/KTO line and one fine-grained baseline (e.g., token/segment-level DPO or span-supervised MT like TWA) under the same data and compute?

Training schedule: Beyond two-stage SFT→DPO, can you try mixed SFT+DPO (interleaved or joint loss) and report its effect vs. your current recipe?

Judge calibration: How sensitive are outcomes to the particular judge and prompt? Any comments on the inter-judge agreement? Any human-correlation study for Ref-Free vs. RefEval/RefMatch?

Multi-reference: (1) I don't see the multi-reference results in Figure 3. (2) Do multi-reference votes always help? What’s the marginal gain of 1→2→3 references? How did you create the multiple references?

Significance: Please add 95% CIs (bootstrap) and a simple permutation test for key deltas in main tables.

---

> ### Author Response · Authors · 2025-11-26
> **Response to Reviewer 2bbc [Part 1]**
>
> We thank you for the detailed review, constructive feedback, and valuable suggestions.
>
> > “Methodologically, the novelty feels incremental: reference-guided evaluation and reference-distilled training have both appeared before, and the paper mostly scales and systematizes them rather than introducing a new objective or learning signal.”
>
> Please see our general response for a detailed discussion: our main contribution is **improving RL-based training for _non-verifiable_ tasks**, where we demonstrate that reference-guided judges enable self-improvement that matches or exceeds reward models without the complexity of training separate RMs or requiring preference data.
>
> ---
> > “The comparisons also underplay strong preference-optimization baselines (e.g., SimPO, ORPO, KTO) and fine-grained supervision relevant to credit assignment (token/segment-level DPO; span-supervised MT like TWA).” “Baseline coverage: Can you add at least one strong SimPO/ORPO/KTO line and one fine-grained baseline (e.g., token/segment-level DPO or span-supervised MT like TWA) under the same data and compute?”
>
> We believe the concern arises from a misunderstanding of our scope. Our work focuses on **reference-guided LLMs-as-Judges**, whose purpose is to improve the *supervision signal* used in preference optimization. This contribution is orthogonal to the specific choice of preference-optimization algorithm. Consistent with the framing of the paper, the main empirical question we study is whether **LLM-judges enhanced by high-quality references can lead to better-aligned policies.**
>
> Therefore, our core comparisons are across **different reward models / LLM-judge variants**, not across SimPO/ORPO/KTO or other optimization algorithms. We use DPO as the optimization method because it is widely adopted and provides a stable setup for isolating the effect of the reward model. All training experiments use DPO under identical settings, making the comparisons fair: the only factor that varies is the LLM-judge / reward model.
>
> Adding SimPO, ORPO, KTO, token/segment-level DPO, or span-supervised MT baselines would mix two separate dimensions: differences in optimization algorithms and differences in supervision quality. Since our goal is to evaluate how our reference-based protocol affects LLM-judge behavior, varying the optimization algorithm would confound the comparison and make it difficult to draw meaningful conclusions about the supervision signal itself. Such baselines answer a different question and fall outside the scope of this paper.
>
> ---
> > “Robustness is not fully stress-tested: results rely on high-quality references, but we don’t see sensitivity to noisy, weaker, or stylistically different references, nor clear evidence against style coupling to the reference generator.” “Reference robustness: How do results change if you paraphrase, shorten, or inject minor errors into references, or use weaker LLMs as references? Any curve of judge accuracy vs. reference quality?”
>
> Please refer to our results in General Response (Point 1). We show that using references from much smaller models (e.g., Nemo-12B) still yields improvements over reference-free baselines, demonstrating robustness to noise and lower model capability.

---

> ### Author Response · Authors · 2025-11-26
> **Response to Reviewer 2bbc [Part 2]**
>
> > “statistical reporting is light (few CIs/permutation tests)” “Significance: Please add 95% CIs (bootstrap) and a simple permutation test for key deltas in main tables.”
>
> Thank you for the suggestion. We have conducted statistical tests accordingly.
> 1. **Main results of Table 1 with 95% bootstrap confidence intervals in the manuscript: the average performance of different evaluation methods**
> | Method | Avg |
> | :--- | :--- |
> | LLMBar-Base | 72.3 (-1.4, +1.5) |
> | HREF-Base | 72.5 (-1.5, +1.6) |
> | CoT | 71.2 (-1.5, +1.6) |
> | Prepair | 74.0 (-1.3, +1.4) |
> | Self-Ref | 73.3 (-1.3, +1.3) |
> | Self-Metric-Ref | 74.6 (-1.3, +1.4) |
> | Ref-Free (Ours) | 73.7 (-1.3, +1.4) |
> | LLMBar-Ref | 74.0 (-1.3, +1.4) |
> | HREF-Ref | 74.8 (-1.3, +1.4) |
> | RefMatch | 77.7 (-1.6, +1.6) |
> | RefEval | **79.1 (-1.4, +1.5)** |
>
> We found that RefEval achieves the highest average accuracy. RefEval significantly outperforms all baseline methods (p < 0.05 on Avg); RefMatch is the second-best method, also outperforming baselines but is significantly worse than RefEval on 4 of 6 columns. Both RefEval and RefMatch substantially improve over reference-free baselines.
>
> 2. **Results for Table 2 in the manuscript (per-model performance)**
> | Method | qwen-2.5-72b | llama-3.1-70b | gemma-2-27b | qwen-2.5-14b | mistral-nemo | gemma-2-9b | glm-4-9b | llama-3.1-8b | qwen-2.5-7b | llama-3-8b | mistral-7b-v0.3 |
> | --- | --- | --- | --- | --- | --- | --- | --- | --- | --- | --- | --- |
> | LLMBar-Base | 79.4 (-1.9, +1.8) | 85.2 (-1.8, +1.7) | 82.3 (-2.1, +2.0) | 81.5 (-1.9, +1.9) | 65.6 (-2.2, +2.4) | 80.8 (-1.8, +1.9) | 71.8 (-1.8, +2.1) | 65.0 (-2.4, +2.3) | 73.5 (-1.9, +2.1) | 60.1 (-2.4, +2.5) | 47.0 (-2.6, +2.6) |
> | Ref-Free (Ours) | 83.4 (-1.9, +1.8) | 85.6 (-1.7, +1.8) | 80.1 (-1.9, +2.0) | 83.3 (-1.8, +1.9) | 63.7 (-2.2, +2.3) | 80.8 (-2.2, +2.0) | 77.3 (-2.0, +2.0) | 71.8 (-2.1, +2.1) | 74.5 (-2.0, +2.1) | 72.3 (-2.1, +2.1) | 61.2 (-2.3, +2.4) |
> | RefEval | **84.6 (-1.8, +1.8)** | **85.9 (-1.7, +1.8)** | **84.9 (-1.7, +1.8)** | **82.4 (-2.2, +2.1)** | **73.2 (-2.1, +2.2)** | **85.7 (-2.0, +2.0)** | **79.5 (-1.9, +1.9)** | **79.4 (-2.2, +2.2)** | **77.4 (-2.0, +2.0)** | **77.5 (-2.1, +2.3)** | **69.6 (-2.2, +2.4)** |
>
> We found that RefEval significantly outperforms both LLMBar-Base and Ref-Free (our own reference-free baseline)  for all models; improvements are consistent across model sizes (7B to 72B parameters).
>
> 3. We have also added the confidence intervals and conducted a paired bootstrap test for the training experiments (similar to the suggested permutation test) comparing RefFree and RefEval for both Llama-3-8B-Instruct and Qwen2.5-7B-SFT. The results show that RefEval outperforms RefFree with statistical significance ($p < 0.05$).
>
> **Results for reference-guided self-improvement (Table 4)**
> | Method        | Llama-3-8B-Instruct AE | Llama-3-8B-Instruct AH | Qwen2.5-7B-SFT AE | Qwen2.5-7B-SFT AH |
> |---------------|-------------------------|-------------------------|--------------------|--------------------|
> | **Base**          | 25.0 (-0.2, +0.3)        | 27.1 (-1.8, 1.7)         | 14.4 (-0.1, +0.2)   | 23.4 (-2.1, 2.1)    |
> | **ArmoRM-Base**   | 49.2 (-0.5, +0.5)        | 40.4 (-2.4, 2.4)         | 32.6 (-0.3, +0.3)   | 58.6 (-2.3, 2.6)    |
> | **V3-Distill**    | 53.9 (-0.5, +0.6)        | 42.2 (-2.1, 2.3)         | 48.8 (-0.5, +0.5)   | 56.5 (-2.3, 2.1)    |
> | **ROUGE**         | 56.4 (-0.6, +0.6)        | 52.1 (-2.2, 2.5)         | 50.9 (-0.5, +0.5)   | 67.4 (-2.0, 2.5)    |
> | **BERTScore**     | 58.8 (-0.6, +0.6)        | 53.0 (-2.8, 1.9)         | 55.3 (-0.5, +0.6)   | 64.5 (-2.3, 2.6)    |
> | **ArmoRM**        | **73.9** (-0.7, +0.7)    | 58.6 (-0.7, +0.7)        | 66.8 (-0.7, +0.7)   | 72.2 (-2.2, 2.1)    |
> | **RefFree**       | 67.5 (-0.7, +0.7)        | 53.8 (-2.2, 2.6)         | 65.1 (-0.6, +0.7)   | 71.8 (-2.1, 2.1)    |
> | **RefEval**       | 73.1 (-0.7, +0.7)        | **58.7** (-2.8, 2.6)     | **70.0** (-0.7, +0.7) | **74.1** (-2.4, 2.0) |

---

> ### Author Response · Authors · 2025-11-26
> **Response to Reviewer 2bbc [Part 3]**
>
> > “most end metrics are still LLM-judge based; more targeted human spot-checks would help address circularity.”
>
> 1. We’d like to clarify that the evaluations of LLMs-as-Judges are based on **human annotations**. The evaluations of trained models use AlpacaEval and Arena-Hard with GPT-4o, which are standard evaluation methods widely adopted by the community.
>
> 2. We agree that human evaluations of the trained policies can offer additional insights. However, running such evaluations across multiple trained models is very costly, especially since the instructions are complex and the outputs are long and free-form. We therefore rely on automatic benchmarks, AlpacaEval and Arena-Hard, which have become standard practice in the community.
>
> 3. We also want to highlight a design choice that reduces concerns about unfair evaluations introduced by circularity: the references used during training are generated by DeepSeek-v3, whereas the LLM evaluator in AlpacaEval and Arena-Hard is GPT-4o. This separation reduces the risk that the evaluator disproportionately favors models whose outputs resemble its own.
> > “Missing citation and baseline: RevisEval: Improving LLM-as-a-Judge via Response-Adapted References — Qiyuan Zhang, ICLR 2025.”
>
> Please note that we have already cited and discussed this paper on Lines 073 and 126. Please refer to our general response for a detailed comparison, where we show that our proposed evaluation method outperforms RevisEval when using the same LLM.
>
> ---
> > “Training schedule: Beyond two-stage SFT→DPO, can you try mixed SFT+DPO (interleaved or joint loss) and report its effect vs. your current recipe?”
>
> We appreciate the suggestion, but this experiment is not directly informative regarding the effectiveness of our framework or the problem we study. Our work focuses on improving the supervision signal via reference-guided LLMs-as-Judges under a fixed, standard SFT→DPO setup. Exploring mixed or interleaved SFT+DPO schedules is out of scope and would not affect our main conclusions, since the improvements we observe come from the enhanced LLM-judge rather than the training schedule.
>
> ---
> > “Judge calibration: Any comments on the inter-judge agreement? Any human-correlation study for Ref-Free vs. RefEval/RefMatch?”
>
> Thank you for the question. We’d like to note that the evaluation methods (Ref-Free, RefEval, RefMatch) are already evaluated on **human-annotated datasets** (Table 1 & 2).
>
> Regarding your question, we have computed the average pairwise agreement between 11 open-source judge models with Ref-Free and RefEval methods, averaging across all datasets:
>
>
>   | Group | Ref-Free | RefEval | Difference      |
>   |-------|----------|---------|--------|
>   | Nat   | 80.98%   | 85.31%  | +4.33% |
>   | Adv   | 75.59%   | 77.61%  | +2.02% |
>   | MT    | 75.35%   | 82.42%  | +7.07% |
>   | Ins   | 76.95%   | 79.69%  | +2.74% |
>   | HREF  | 74.17%   | 81.83%  | +7.66% |
>   | Avg   | 76.61%   | 81.37%  | +4.76% |
>
>
> The result shows that RefEval helps  **ground** 11 OS LLMs, reducing variance in their decisions and making them more consistent with each other.
>
>
> Furthermore,  we provide agreement with GPT-4o-as-Judge as ground truth for selected models as judges (7b, 27b, 70b):
>
>
>   | Model         | Method   | Nat   | Adv   | MT    | Ins   | HREF  | Avg   |
>   |---------------|----------|-------|-------|-------|-------|-------|-------|
>   | qwen-2.5-7b   | Ref-Free | 84.00 | 78.06 | 78.50 | 72.51 | 75.21 | 77.65 |
>   | qwen-2.5-7b   | RefEval  | 89.00 | 84.33 | 80.50 | 78.59 | 79.72 | 82.43 |
>   | gemma-2-27b   | Ref-Free | 85.00 | 79.00 | 74.00 | 83.70 | 83.94 | 81.13 |
>   | gemma-2-27b   | RefEval  | 92.00 | 86.83 | 86.50 | 84.67 | 86.48 | 87.30 |
>   | llama-3.1-70b | Ref-Free | 87.00 | 83.70 | 89.00 | 87.83 | 93.24 | 88.15 |
>   | llama-3.1-70b | RefEval  | 92.00 | 89.03 | 86.50 | 90.51 | 89.01 | 89.41 |
>   | qwen-2.5-72b  | Ref-Free | 88.00 | 79.31 | 90.50 | 87.35 | 89.58 | 86.95 |
>   | qwen-2.5-72b  | RefEval  | 90.00 | 86.21 | 92.50 | 88.08 | 88.73 | 89.10 |
>
>
> We found that (1) RefEval improves alignment with GPT-4o, (2) larger models already agree well with GPT-4o (~87-89%), leaving less room for improvement, and (3) the difficult set LLMBar_adversarial (Adv) shows the largest consistent improvement (+6.58% avg).
>
> ---
> > “How sensitive are outcomes to the particular judge and prompt?”
>
> Thank you for raising the question. As we noted in Lines 206-209, we explored prompt variants for our core methods RefEval and RefMatch. We detailed those variants and judges in Appendix A and Appendix G. For example, we introduce CoT or add additional judging rules (rubrics) in the prompts. Across these experiments, we found consistent improvement from our reference-guided evaluation method compared to the baselines. Please refer to the manuscript for details. In the revised version, we will make the sensitivity experiments more salient.

---

> ### Author Response · Authors · 2025-11-26
> **Response to Reviewer 2bbc [Part 4]**
>
> > “Multi-reference: (1) I don't see the multi-reference results in Figure 3. (2) Do multi-reference votes always help? What’s the marginal gain of 1→2→3 references? How did you create the multiple references?”
>
> 1. *“I don't see the multi-reference results in Figure 3.”* As we explained in Lines 318-319, the multi-reference result is labeled **Vote**, which is the last value on the x-tick: “this voting approach yields the highest average accuracies for both protocols” for RefEval and RefMatch. Detailed results are provided in Appendix A. We appreciate the question and will make this presentation more salient in the revised manuscript.
>
> 2. *“How did you create the multiple references?”* As noted in Lines 309-311, we sample completions from frontier models for our prompt instances. The models include Claude 3.7, Gemini 2.0, DeepSeek-V3, and others.
>
> 3. *“What’s the marginal gain of 1→2→3 references? …always help?”* Thanks for the question. To directly address the progression of gains (1→2→3 references), we compiled the average accuracy of **RefEval** across 11 open-source judges (matching the experiment in Figure 3) using voting ensembles of different sizes. Voting consistently improves performance, though with diminishing returns. The move from a single reference to a 3-reference vote yields the most substantial robustness gain.
>
> | Method | # Refs | Avg. Accuracy | Delta |
> | :--- | :---: | :---: | :---: |
> | Single Best (Avg) | 1 | 81.4% | - |
> | *Pairwise Ensemble* | *2* | *81.8%* | *+0.4%* |
> | **Majority Vote** | **3** | **82.2%** | **+0.4%** |
> | Majority Vote | 5 | 82.3% | +0.1% |
>
> **Note**: "1 Ref" is the average of top single models (Gemini-2.5, GPT-4.1, Claude-3.7). "2 Refs" is a simulated ensemble using tie-breaking. "3" and "5" are the actual majority voting results. We want to note that multiple references will introduce additional cost.
>
> The detailed full table is as follows:
>
> *(Results averaged across 11 open-source judges on 5 datasets)*
> | Method | Ref Source(s) | Nat | Adv | MT | Ins | HREF | **Avg** |
> | :--- | :--- | :--- | :--- | :--- | :--- | :--- | :--- |
> | **Single Ref** | **Gemini-2.5-Flash** | 0.874 | 0.724 | 0.755 | 0.750 | 0.882 | **0.818** |
> | | **GPT-4.1** | 0.891 | 0.742 | 0.764 | 0.751 | 0.884 | **0.815** |
> | | **Claude-3.7-Sonnet** | 0.885 | 0.710 | 0.765 | 0.754 | 0.859 | **0.809** |
> | | **GPT-4o** | 0.868 | 0.749 | 0.767 | 0.745 | 0.898 | **0.807** |
> | | **DeepSeek-V3** | 0.889 | 0.706 | 0.762 | 0.738 | 0.884 | **0.805** |
> | **3-Ref Vote** | **Claude-3.7 + Gemini-2.5 + GPT-4** | 0.893 | 0.737 | 0.775 | 0.763 | 0.897 | **0.822** |
> | | **Claude-3.7 + Gemini-2.5 + GPT-4.1** | 0.894 | 0.735 | 0.765 | 0.761 | 0.893 | **0.821** |
> | | **Claude-3.7 + Gemini-2.5 + V3** | 0.894 | 0.723 | 0.763 | 0.760 | 0.886 | **0.817** |
> | | **Llama-405B + GPT-4.1 + Gemini-2.5** | 0.895 | 0.751 | 0.765 | 0.769 | 0.893 | **0.835** |
> | **5-Ref Vote** | **Gem2 + Gem2.5 + 4.1 + V3 + Cla3.7** | 0.898 | 0.739 | 0.767 | 0.776 | 0.876 | **0.823** |

---

### Official Review · Reviewer_yDFd · 2025-11-03

**Soundness:** 3
**Presentation:** 3
**Contribution:** 2
**Rating:** 4
**Confidence:** 3

**Summary:**

This paper investigates whether high-quality reference outputs can improve LLM alignment through enhanced evaluation and training methods. The authors develop reference-guided LLM-as-a-Judge protocols (RefEval and RefMatch) and demonstrate their effectiveness in both evaluation accuracy and self-improvement training scenarios.

**Strengths:**

* Evaluation across diverse datasets (Natural, Adversarial, MTBench, Instrusum, HREF, AlpacaEval and ArenaHard) and models (Qwen, Llama, GPT, Gemma, GLM, etc)
* Useful settings and approaches. RefEval provides explicit guidance on using reference outputs as benchmarks for instruction-following quality. RefMatch provides a good baseline to compare.

**Weaknesses:**

I mostly concern experiment design.
* Missing comparison to recent reference-based methods like RevisEval beyond a brief mention
* No systematic study of how reference quality affects downstream performance
* Unclear how to obtain high-quality references in domains where frontier models struggle
* No analysis of how reference diversity affects evaluation robustness, especially for aspects that reference responses miss.

**Questions:**

N/A

---

> ### Author Response · Authors · 2025-11-26
> **Response to Reviewer yDFd**
>
> Thank you for your review and for acknowledging that our evaluation covers diverse datasets and models.
>
> > “Missing comparison to recent reference-based methods like RevisEval beyond a brief mention”
>
> As detailed in our general response (Part 2, Point 3), RevisEval and our work serve different goals. RevisEval is an evaluation-only technique that generates response-adapted references. Our work is an **alignment training** framework and not a mere evaluation improvement attempt. We utilize high-quality references not just to score, but to drive a model self-improvement (Section 5). We show that static, high-quality references are sufficient to drive significant training gains, a scope RevisEval does not cover.
>
> Furthermore, we propose an evaluation method that **outperforms theirs in comparable settings**: their fine-tuned judge from Llama-3.1-8B-Instruct achieves 64.9 on LLMBar on average, while our method RefEval yields 77.7 with non-finetuned Llama-3.1-8B-Instruct as the judge under GPT4o-generated reference, which is a significant improvement. We’d like to provide further results when other frontier models are used as the reference source:
>
>
> | Method ( Llama-3.1-8B as Judge)   | llmbar_nat | llmbar_adv | Avg Acc.   |
> |---------------------------------|------------|------------|-------|
> | RefEval-GPT4o | 0.810 | 0.745 | 0.777 |
> | RefEval-GPT-4.1                 | 0.850      | 0.730      | 0.790 |
> | RefEval-Claude-3.7-Sonnet       | 0.855      | 0.716      | 0.786 |
> | RefEval-Gemini-2.5-Flash        | 0.825      | 0.734      | 0.779 |
> | Ref-Free (Ours) | 0.760 | 0.629 | 0.694 |
> | RevisEval (Zhang et al., 2024)  | -          | -          | 0.649 |
>
>
> As shown in the results, our method outperformed RevisEval on the LLMBar dataset using the same judge model.
>
> ---
> > “No systematic study of how reference quality affects downstream performance”
>
> Please see our General Response (Point 1), where we provide a detailed discussion. We will include these results in the revised manuscript. The results show that while stronger references help more, even weaker ones (e.g., Nemo-12B) still improve over the reference-free baseline. Moreover, in the self-improvement training setting, using references from a less capable LLM (gpt-4o-mini) also yields improvements over the reference-free baseline.
>
> ---
> > “Unclear how to obtain high-quality references in domains where frontier models struggle”
>
> We agree that obtaining high-quality references can be especially challenging in complex domains. However, this is a fundamental challenge for all supervised learning, not specific to our method. If a domain is so difficult that no references (human or model) exist, then even standard SFT or RLVR cannot be performed. Our contribution is to show that when SFT is possible (i.e., references exist), our method extracts more value from those references than SFT alone. In this sense, our method is analogous to RLVR – RLVR for domains such as math reasoning requires ground-truth answers, while our method requires high-quality references for non-verifiable tasks and general alignment.
>
> ---
> > “No analysis of how reference diversity affects evaluation robustness, especially for aspects that reference responses miss.”
>
> Our original manuscript has already conducted a study (Figure 3) showing that references generated by LLMs from different model families (e.g., GPT-4o, Claude-3.5, Gemini-2.0) can substantially improve the LLM-judge’s performance, indicating that **our evaluation method generalizes to references with different characteristics**.
>
> We have also explored Multi-Reference strategies (Line 316), in Appendix A.5 of our original manuscript. Please refer to Table 10, where we test “Multi-Ref Avg” and “Multi-Ref Voting.” We find that while multi-reference voting yields the highest accuracy (Figure 3, labeled “Vote” on the x-axis), a single high-quality reference (RefEval) captures most of the performance gain and offers the best efficiency-accuracy trade-off.

---

### Author Response · Authors · 2025-11-26
**General Response [Part1]**

We thank the reviewers for their detailed feedback and constructive comments, and for recognizing the strengths of our work, including the "excellent" soundness and contribution (Reviewer r7ga), the "broad and systematic evaluation" (Reviewer P3EF), and the "practical recipe" for alignment (Reviewer 2bbc).

We address three common concerns raised by several reviewers below.


**1. On Reference Quality and Robustness (Reviewers yDFd, 2bbc, P3EF)**

Several reviewers raise concerns about the dependence of our method on high-quality references and what happens when they are weaker or noisy. We'd like to respond:

(1) **Impact of reference quality on LLM-judge’s performance**. We have conducted an ablation study (please see the table below) using references generated by smaller, less capable models (Mistral-Nemo-12B and Tulu-2-7B) instead of GPT-4o. We tested this with Llama-3.1-70B as the judge across four datasets (Nat, Adv, MT, Ins). The results show that while reference-aided evaluation still outperforms reference-free evaluation, using stronger references yields better performance.
| Judge Model    | Reference Source | Method | Avg. Acc. |
|----------------|------------------|--------|-----------|
| Llama-3.1-70B  | GPT-4o           | RefEval| 0.856     |
|                | Nemo-12B         | RefEval| 0.850     |
|                | Tulu-7B          | RefEval| 0.834     |
|                | None         | RefFree | 0.832    |


(2) **Impact of reference quality on LLM self-improvement**: We conducted an additional set of training experiments following the same setup as Section 4 of our paper, but using reference outputs generated by GPT-4o-mini, a weaker model compared to DeepSeek-V3, which was used to produce references in the original experiments. These experiments are based on Llama-3-8B-Instruct, where the model is first fine-tuned via SFT (distillation), then undergoes self-improvement with and without references. The results show that: (1) the trends observed in the original experiments generalize to weaker references – models can still self-improve beyond SFT, and references continue to enhance this process; and (2) the quality of references impacts the effectiveness of reference-guided self-improvement, with models trained using DeepSeek-V3 references outperforming those trained with GPT-4o-mini references.


| Method                             | AlpacaEval | Arena-Hard |
|------------------------------------|-----------:|-----------:|
| Base (Llama-3-8B-Instruct)         |       25.0 |       27.1 |
| DeepSeek-V3                        |       84.8 |       94.9 |
| DeepSeek-V3 Distill                |       53.9 |       42.2 |
| DeepSeek-V3 RefFree                |       67.5 |       53.8 |
| DeepSeek-V3 RefEval                |       73.1 |       58.7 |
| GPT-4o-mini                        |       49.2 |       74.5 |
| GPT-4o-mini Distill                |       28.7 |       40.7 |
| GPT-4o-mini RefFree                |       42.6 |       41.7 |
| GPT-4o-mini RefEval                |       44.4 |       58.3 |


(3) **Analysis of how reference diversity affects evaluation robustness**: our original manuscript has already conducted a study (Figure 2) showing that references generated by LLMs from different model families (e.g., GPT-4o, Claude-3.5, Gemini-2.0) can substantially improve the LLM-judge’s performance, which indicates that our evaluation method generalizes to references of different characteristics.


(4) We’d like to note that the reference-guided training paradigm we study **assumes access to high-quality reference outputs**. This assumption is not unique to our method, and in fact, standard SFT and RLVR both are essentially using high-quality reference outputs (i.e., in SFT the models are fine-tuned on the reference outputs and in RLVR, reference outputs are used to provide the reward signal). Therefore, dependence on reference quality is a **standard design choice**, not a specific limitation of our approach. While systematically studying how reference quality affects training effectiveness is a meaningful next step, our focus here is on establishing and evaluating the training paradigm itself for non-verifiable tasks.

---

### Author Response · Authors · 2025-11-26
**General Response [Part2]**

2. **Regarding concerns of novelty of our work: “reference-guided evaluation and reference-distilled training have both appeared (Reviewer 2bbc)”; “the core concept of reference-guided judging remains closely related to prior work (Reviewer P3EF)”**

   (a). We’d like to note that our primary contribution is **improving RL-based training for __non-verifiable__ tasks** (or general alignment tuning), which, to our knowledge, hasn't been systematically studied before. Our studied training paradigm narrows the gap between RLVR (verifiable rewards) and non-verifiable domains by addressing a key limitation: outputs in non-verifiable domains cannot be automatically checked against a reference or gold-standard answer. To the best of our knowledge, we are the first to demonstrate the strong effectiveness of **reference-guided alignment tuning**. We provide a systematic recipe showing that, given high-quality SFT data, reference-guided LLMs-as-Judges can drive self-improvement **without an external reward model**, achieving performance comparable to or better than training with finetuned reward models.

    (b). For example, we show that Llama-3.1-8B-Instruct can self-improve using reference outputs generated by DeepSeek V3, reaching 73.1 and 58.7 on AlpacaEval and Arena-Hard. These gains substantially exceed prior SOTA models trained from the same base model. As one reference point, the SimPO model trained with an external reward model (ArmoRM) only achieves 51.6 and 36.2 on these benchmarks (Table 2).

    (c). Moreover, our proposed evaluation method outperformed previous methods that use references to enhance LLMs-as-Judges. Below, we provide a detailed comparison with a closely related recent work.

---
3. **Comparison to RevisEval (Zhang et al., 2024). (Reviewers yDFd, 2bbc)**

   Reviewers have asked for comparisons with RevisEval. While we have already discussed RevisEval in our original manuscript, we provide a detailed comparison below:

   RevisEval focuses on modifying references to match the candidate's response length/style strictly for evaluation accuracy. In contrast, our work focuses on alignment training. We use high-quality references to guide self-improvement. RevisEval does not explore this training pipeline and only focuses on static evaluation settings. Furthermore, our proposed evaluation method outperforms theirs in comparable settings: their fine-tuned judge from Llama-3.1-8B-Instruct achieves 64.9 on LLMBar, while our method RefEval yields 77.7 with non-finetuned Llama-3.1-8B-Instruct as the judge under GPT4o-generated reference. We’d like to provide further results when other frontier models are used as reference sources:

   | Method ( Llama-3.1-8B as Judge)   | llmbar_nat | llmbar_adv | Avg Acc.   |
   |---------------------------------|------------|------------|-------|
   | RefEval-GPT4o | 0.810 | 0.745 | 0.777 |
   | RefEval-GPT-4.1                 | 0.850      | 0.730      | 0.790 |
   | RefEval-Claude-3.7-Sonnet       | 0.855      | 0.716      | 0.786 |
   | RefEval-Gemini-2.5-Flash        | 0.825      | 0.734      | 0.779 |
   | Ref-Free (Ours) | 0.760 | 0.629 | 0.694 |
   | RevisEval (Zhang et al., 2024)  | -          | -          | 0.649 |

   As shown in the results, our method outperformed RevisEval (Zhang et al. 2024) on the LLMBar dataset using the same judge model.

---

### Meta-Review · Area_Chair_UWR7 · 2025-12-28

**Summary:**

This paper investigates whether high-quality reference outputs can improve LLM alignment through enhanced evaluation and training methods. The authors develop reference-guided LLM-as-a-Judge protocols (RefEval and RefMatch) and demonstrate their effectiveness in both evaluation accuracy and self-improvement training scenarios.

**Reviewer Concerns:**

Strengths:
1. The proposed approach is a good supplement to RLVR for non-verifiable domains, which makes this work interesting to the broad community.

2. Comprehensive evaluation across diverse datasets (Natural, Adversarial, MTBench, Instrusum, HREF, AlpacaEval and ArenaHard) and models (Qwen, Llama, GPT, Gemma, GLM, etc).

3. Useful settings and approaches. RefEval provides explicit guidance on using reference outputs as benchmarks for instruction-following quality. RefMatch provides a good baseline to compare.

4. Good results and insightful analysis.

Weaknesses: (authors provided good responses to reviewers' major concerns)

1. Missing comparison to recent reference-based methods like RevisEval beyond a brief mention. (already provided in the rebuttal)
2. No systematic study of how reference quality affects downstream performance or does not provide clear criteria or measurements for selecting or validating high-quality references. (Mostly addressed in the rebuttal)
3. No analysis of how reference diversity affects evaluation robustness, especially for aspects that reference responses miss.

**Reviewer Scores:**

Review ratings come out as 2, 4, 4, 10, while the 10 is obviously over-rated. The authors provide fairly good responses which address most of the concerns.

---

### Decision · Program_Chairs · 2026-01-26

Accept (Poster)